# Precision-induced localized molten liquid metal stamps for damage-free transfer printing of ultrathin membranes and 3D objects

Chuanqian Shi[1,2,3], Jing Jiang[3], Chenglong Li [3], Chenhong Chen[3], Wei Jian[4] & Jizhou Song [3,5,6] ✉

Transfer printing, a crucial technique for heterogeneous integration, has gained attention for enabling unconventional layouts and high-performance electronic systems. Elastomer stamps are typically used for transfer printing, where localized heating for elastomer stamp can effectively control the transfer process. A key challenge is the potential damage to ultrathin membranes from the contact force of elastic stamps, especially with fragile inorganic nanomembranes. Herein, we present a precision-induced localized molten technique that employs either laser-induced transient heating or hotplate-induced directional heating to precisely melt solid gallium (Ga). By leveraging the fluidity of localized molten Ga, which provides gentle contact force and exceptional conformal adaptability, this technique avoids damage to fragile thin films and improves operational reliability compared to fully liquefied Ga stamps. Furthermore, the phase transition of Ga provides a reversible adhesion with high adhesion switchability. Once solidified, the Ga stamp hardens and securely adheres to the micro/nano-membrane during the pick-up process. The solidified stamp also exhibits the capability to maneuver arbitrarily shaped objects by generating a substantial grip force through the interlocking effects. Such a robust, damage-free, simply operable protocol illustrates its promising capabilities in transfer printing diverse ultrathin membranes and objects on complex surfaces for developing high-performance unconventional electronics.

Soft/stretchable integrated electronic systems with unconventional desired layouts show superior mechanical compliance and deformability and thus can be applied in unusual places that are not possible for conventional rigid electronics, such as bio-inspired cambered imagers[1–3], wearable/implantable health devices[4–8], human-machine interactions[9–11], and deformable displays[12–14]. To achieve electronic performances comparable to the commercial hard devices fabricated by printed circuit boards (PCBs), soft/stretchable electronics usually

[1]Center for Mechanics Plus under Extreme Environments, School of Mechanical Engineering & Mechanics, Ningbo University, Ningbo, China. [2]Key Laboratory of Impact and Safety Engineering, Ministry of Education, Ningbo University, Ningbo, China. [3]Department of Engineering Mechanics, Soft Matter Research Center, Key Laboratory of Soft Machines and Smart Devices of Zhejiang Province, State Key Laboratory of Brain-Machine Intelligence, Zhejiang University, Hangzhou, China. [4]Zhejiang-Italy Joint Lab for Smart Materials and Advanced Structures, School of Mechanical Engineering & Mechanics, Ningbo University, Ningbo, China. [5]Department of Rehabilitation Medicine, The First Affiliated Hospital School of Medicine, Zhejiang University, Hangzhou, China. [6]Institute of Flexible Electronics Technology of THU, Zhejiang, Jiaxing, China. ✉e-mail: jzsong@zju.edu.cn

require fabricating micro/nano-scale inorganic materials (metal and semiconductor)[15,16] with soft substrates. However, fragile micro/nano-thick inorganic membranes are commonly prepared on a hard wafer and need to be integrated onto soft substrates in a controllable, damage-free, and deterministic fashion through heterogeneous integration technologies. A promising heterogeneous integration technology is transfer printing, which proves an emerging manufacturing technique[17,18] with the ability to use a stamp to transfer print inks (e.g., inorganic materials) onto flat or even complicated curved substrates' surfaces[19–22].

The transfer printing process is to mediate the adhesion between two interfaces of three layers (stamp/ink/substrate), and the adhesion of the stamp/ink interface should be stronger (or weaker) than the ink/substrate interface during the pickup (or printing) process[23,24]. The controllability and functionality of reversible adhesion may be enhanced by methods such as chemistry bond assisted[25,26], kinetically controlled[27–30], laser non-contact driven[31–34], temperature-responsive[19,35–37], magnets-driven[38], and water assisted[25,39,40]. The widely used transfer printing stamps are soft stamps, with poly-dimethylsiloxane (PDMS) being the most commonly used. The PDMS stamps take advantage of the viscoelastic nature to control adhesion during the pickup and printing process[27,41,42]. Other soft materials such as shape memory polymers (SMPs)[31,35,43], water response hydrogels[44,45], the latex balloon[20,21], and thermal release tapes[19,46,47] have also been developed as ideal candidates for stamps. Furthermore, microscale surface reliefs on soft stamps[29,33,48], such as line and space geometry, pyramids, and micro-tips, were developed to provide high adhesion switchability between the pickup and printing process. Despite the notable advances in the above solid-based polymeric stamps, cracks and wrinkles often occur in inorganic thin films during the transfer printing process, whether on flat or curved surfaces. This is primarily due to the shear stresses induced by the compression of the soft stamps at the stamp/ink/substrate interfaces[40,49].

Other transfer printing stamps are based on liquid-phase materials, which take advantage of the fluidity of liquid to avoid damage when contacting inorganic thin films and allow complete transfer printing of thin films from hard wafers to complex, specially shaped interfaces[39,40]. However, the adhesion force of a liquid stamp is developed by Laplace pressure and surface tension, which is much weaker than the adhesion of a soft stamp and not applicable to pick up the inks with heavy mass, complex 3D profiles, or high adhesion strength between ink/substrate interfaces. Therefore, there is a strong need for a technique with high revisable adhesion that can deal with the stress mismatch from the contact process and yet have a high switchable adhesion in the pickup and printing processes. The liquid-solid phase change materials obtain reversible adhesive characteristics, which could help meet the challenges[36,50,51]. On the basis of the phase change of metal gallium, it has reported a highly reversible and switchable adhesion, demonstrating the feasibility of using gallium's phase change for strong, reversible, and robust adhesion on various surfaces under different conditions. This approach has been effectively applied to macroscopic objects[50]. Previous studies also provide critical insights into the behavior of gallium in transfer printing processes, such as the crystallization kinetics of gallium under varying thermal conditions, the liquid crystal structure of supercooled gallium, and the mechanical stability of gallium-oxide nanofilms encapsulating liquid gallium[52–54]. Building on these findings, further research is needed to explore its application for transferring micro/nano-scale objects and delicate ultrathin films due to challenges such as difficult visual observation, small mass, and fragility of thin films. Additionally, completely melted gallium droplets pose operational challenges due to their high fluidity and tendency to easily fall. Therefore, a comprehensive analysis of the critical parameters influencing transfer accuracy and uniformity is essential. Inspired by the precision control achieved through localized heating in elastic stamp transfer printing[31–33], which allows for controllable deformation, we adapted this technique for use with liquid metals. Unlike elastomer stamps, the fluid nature of liquid metals introduces challenges that require careful investigation of localized melting. This includes further investigating the effects of the extent of gallium melting and the curvature of the melting stamp on the precision of the transfer printing process. Notably, previous studies have shown that elastomer stamps can achieve wafer-scale patterning over larger surface areas[19,47]. However, arrays transferred using a globally molten stamp exhibit significant dislocations and extensive gallium residue due to the fluidity of the molten metal. Addressing these challenges is imperative for achieving higher precision and cleaner transfers of ultrathin films when utilizing phase change materials, thereby extending the applicability of metal gallium for transfer printing.

Here in this work, we report a precision-induced (laser or hot-plate) localized molten technique (PLMT) enabled by the Ga stamp (Fig. 1A), which leverages the fluidity of localized molten liquid metal to avoid damage on fragile thin films and utilizes the solid part of the stamp to maintain structural integrity and improve operation reliability. Employing a nearly planar gallium stamp achieved through localized melting can significantly improve the transfer accuracy of thin film arrays and reduce gallium droplet residue compared to a fully molten droplet. Additionally, by adjusting laser power and spot size, the melting range, optimizing the transfer process can be precisely controlled. The simple yet robust thermal actuated, laser-transient-induced or hotplate-directional-induced local molten Ga stamp can provide gentle preload, intimate contact, and highly reversible adhesion strength. Systematic experimental and theoretical studies were carried out to analyze the liquid profile when a liquid Ga droplet approached inks, and quantitatively compare the contact force and adhesion strength between liquid metal stamp and elastic PDMS stamp. As shown in the insets in Fig. 1A, we demonstrate this damage-free, easily operable, highly reversible, and switchable adhesion strategy for performance lossless transfer printing micro/nano-functional membranes in physiological monitoring and circuit systems to enrich the heterogeneous integration capabilities. Additionally, we show that the PLMT strategy can maneuver arbitrarily shaped objects as a universal gripper by intimate contact in a liquid state and lock the target objects in a solid state.

## Results
### Laser-induced transient localized molten technique
Figure 1 shows the laser-transient-induced localized molten technique. As shown in Fig. 1B a metal Ga stamp in a chamber turns to a liquid droplet above 29.76 °C under an outer heating stimulus to approach micro/nano-membranes on a donor substrate. Benefiting from the fluidity of liquid droplets without high shear force, the liquid stamp can have intimate contact with the membranes with barely inducing damage on them under a small preload (Fig. 1C). To switch to a high adhesion state, we cooled the liquid Ga to room temperature (-20 °C) by removing the outer heating stimulus to solidify the stamp. The solidified Ga with the modulus of 9.3 GPa[50] has a tough connection to the micro/nano-thick membranes and it is hard to deform during the retracting process, which yields a high adhesion state of Ga stamp to the ink and ensures a reliable pickup (Fig. 1D). Before the print process, the adhesive force turns to a low state by the localized molten surface of the stamp with a near-infrared laser (808 nm, 13 W) transiently induced to the interface of stamp/ink in 200 ms (Fig. 1E and Supplementary Fig. 1). Supplementary Fig. 1A demonstrates the high-speed photographs and thermograms of the metal gallium stamp before and after laser heating from the side view. The temperature of solid Ga was 20.3 °C before heating and it increased to 36.8 °C when the laser worked on the surface of Ga for ~200 ms. Due to the high laser power, acting on the surface of the gallium metal with a power density of 0.82 W/mm², the surface of the solid gallium stamp rapidly began to

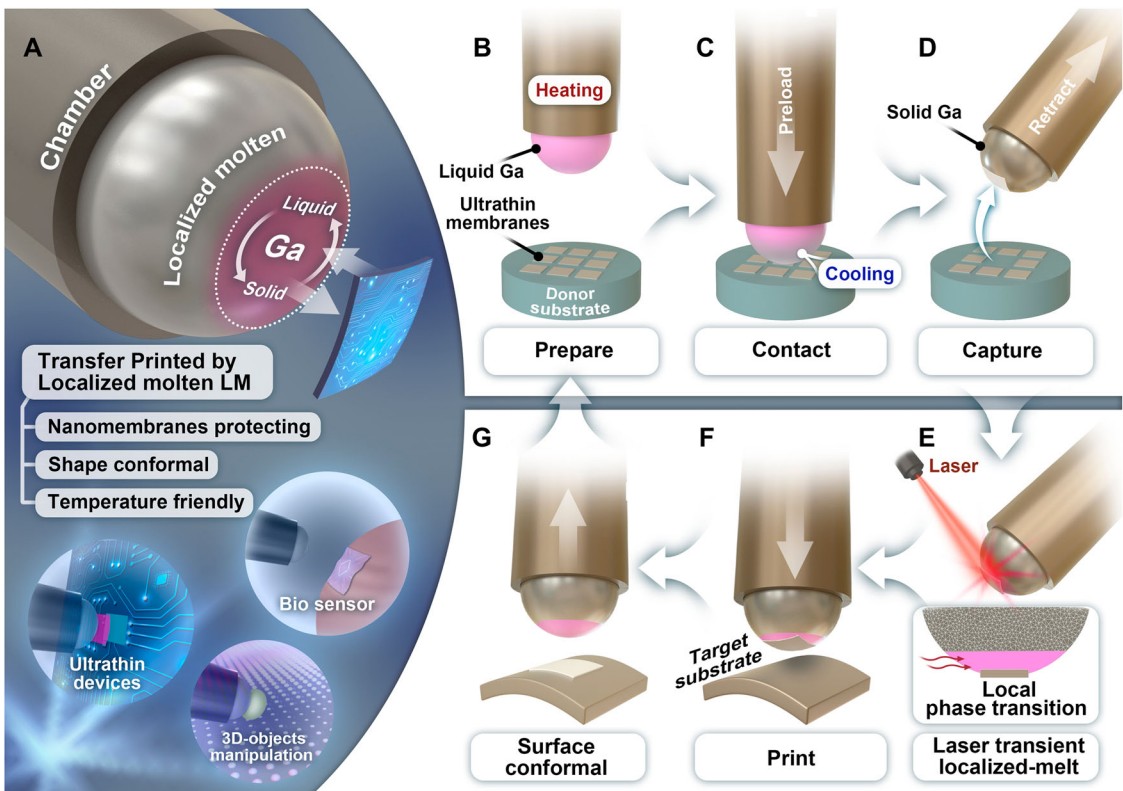

**Fig. 1 | Schematic illustration of the pickup and printing process of PLMT.**
**A** Schematic illustration of the precision-induced localized molten technique and its application prospects. (**B** to **D**) Pickup process. **B** The liquid gallium (Ga) stamp approaches the membranes on the donor substrate. **C** The liquid Ga stamp presses against the membranes, facilitating intimate contact without causing damage. **D** The liquid Ga experiences a phase transition to the high adhesion solid state at

room temperature and then it is retracted. **E**–**G** Printing process. **E** The solid Ga stamp is localized molten by the external laser-transient-induced stimulus. **F** The localized molten Ga with the membrane moves to contact with the curved target substrate. **G** The stamp is removed and completely melted for reuse with the membrane placed onto the target substrate.

melt under the heating of the laser. Then it turned to 20.3 °C after the laser off, and gallium metal still maintained a localized molten state. Supplementary Fig. 1B shows the schematic of a COMSOL®[1] finite element model consisting of a 2 μm-thick Si platelet with a diameter of 400 μm and a Ga stamp with a diameter of 2.3 mm and a height of 1.85 mm. From this simulation, one can see that the temperatures reached in the system are about 36.2 °C, slightly higher than the melting point of metal Ga 29.76 °C, sufficient to cause localized melt without damaging the ink (simulation details in Supplementary Note 3: laser heating temperature gradient analysis). Supplementary Fig. 1C shows the infrared imaging temperature curves of laser heating and cooling for 6 times. When the laser is turned on, the temperature of metal Ga is higher than 30 °C, and it localized melts (Supplementary Fig. 1D and Supplementary Movie 1). The infrared temperature of the Ga stamp drops rapidly after the laser is off, but localized liquefied Ga still takes some time to solidify, leaving operating time for printing. The state of Ga after laser-induced heating resembles a coexistence of liquid and solid phases due to the limited area excited by the laser on the crystal gallium (Supplementary Fig. 1A top row, Supplementary Fig. 1D right, Supplementary Fig. 2A, and Supplementary Fig. 3Aii)[55,56]. Thus, the localized molten stamp, having a low adhesion state at the interfaces, facilitates the printing of inks onto complex curved surfaces without causing any damage to ultrathin films or any difficulty in operations (Fig. 1F). Due to easy operation, the localized molten stamp can be easily retracted (Fig. 1G), and it can be reused in the next process after complete melting.

The localized molten Ga stamp can also help to reduce the dislocation and the pollution of Ga stamp caused by the drop of liquid metal. To validate the efficiency of PLMT, we compared the differences

between localized and global molten Ga stamp transfer printing methods. The experimental results are presented in Supplementary Fig. 3. A 2 × 2 square silicon platelet array (each with dimensions of 200 μm × 200 μm × 2 μm) was initially picked up by a planar solid gallium metal with an infinite radius of curvature (Supplementary Fig. 3Ai). Subsequently, the array was transferred using both a localized molten gallium stamp, which exhibits a slightly reduced radius of curvature compared to a perfectly planar surface and only undergoes surface flow (Supplementary Fig. 3Aii), and a global molten gallium stamp, which forms a single droplet with a radius of curvature decreasing to that of the liquid droplet (~2 mm) (Supplementary Fig. 3A(iii)). When printing the array onto a PDMS substrate using the localized molten stamp, the array remained highly regular, with minimal gallium residue around it (Supplementary Fig. 3B, right with a red solid line box). In contrast, the array transferred with the global molten stamp showed significant dislocations and extensive gallium residue (Supplementary Fig. 3B, left with a blue solid line box). Therefore, localized transfer printing achieves higher precision with less contamination compared to global transfer printing. The increased deformation occurs because a droplet with a smaller radius of curvature needs to transition from a curved surface to a flat substrate upon contact. This transition involves greater deformation compared to a localized molten stamp with a larger radius of curvature approaching flatness. Ideally, ensuring planar contact between a flat stamp and the substrate can maintain transfer precision and prevent dislocation during the printing process.

The residual problem refers to the tendency of liquid metal to remain on the inks during transfer printing, potentially contaminating the receiver substrate. This contamination can affect the ink's

operational lifespan, particularly when using fully molten gallium droplets, which are prone to dropping and polluting the device. To address this, we employ the localized melting method, which effectively minimizes residual issues. Although the residual problem is significantly reduced, minor contamination of gallium metal on printed substrates still exists due to the wrinkle and rupture of the gallium oxide nanofilm under high preload[54]. Here, the remaining gallium can be effectively removed using alcohol as demonstrated in the previous study due to the strong interactions between gallium oxide and oxhydryl[51]. Therefore, we quantified the extent of gallium metal contamination on devices and evaluated its impact on performance. We selected micro-LEDs (with dimensions of 600 μm × 200 μm × 2 μm) as test subjects and examined the gallium residue on the micro-LEDs under different preload conditions. Supplementary Fig. 4A shows photos of liquid gallium stamps contacting micro-LEDs under five different preload conditions. As the preload increases from 1 mN to 5 mN, the gallium metal is gradually compressed, almost flattening under 5 mN preload. Supplementary Fig. 4B displays the gallium residue under a microscope, marked with red dotted lines. Initially, we photographed a micro-LED before Ga contact (Supplementary Fig. 4Bi), where one micro-LED is divided into left and right parts due to the high magnification of the microscope. The area ratio of gallium residue increases from 0.1% (1 mN preload) to 0.66% (5 mN preload). Generally, the residue is minimal and mainly located at the micro-LED edges, with a small amount on the gold electrode pads, possibly due to gallium oxide breaking at sharp edges. Finally, we removed the residual Ga with alcohol (Supplementary Fig. 4B(vii)). To test the impact of gallium contamination on device performance, we selected clean, alcohol-cleaned, and micro-LEDs under 5 mN preload micro-LEDs and tested their performance. Supplementary Fig. 5 shows the light-emitting capability of both clean (Supplementary Fig. 5A), alcohol-cleaned (Supplementary Fig. 5B) and contaminated under 5 mN preload (Supplementary Fig. 5C) micro-LEDs in proper function. Additionally, we provided voltage-current curves for both sets of devices during continuous operation. The results indicate that clean and alcohol-cleaned micro-LEDs functioned normally for 20 days (Supplementary Fig. 5D, left and middle), whereas contaminated micro-LEDs signal was abnormal after 8 days (Supplementary Fig. 5D, right). The anomaly is likely because gallium atoms exhibit embrittlement with the gold electrode pads[57–59]. The cleaned devices have the same lifespan as uncontaminated devices, which indicates that the method of cleaning the contamination with alcohol is very effective in maintaining the good working performance of the transfer printed device.

Next, we discuss three issues in the use of gallium metal stamps. The first is a layer of gallium oxide ($Ga_2O_3$) surrounding the surface of liquid Ga (Supplementary Fig. 6A) in the presence of air, which maintains its rounded shape and aids in the easy peeling of liquid metal from various surfaces (Supplementary Figs. 2B and 6B)[50,60]. Generally, the gallium oxide thickness is kept at 1~3 nm because the gallium oxide skin is localized passivating, which protects the liquid gallium from further oxidizing[61]. However, gallium oxide may break in the preload process when in contact with irregular or sharply angled objects due to its non-stretchable nature (Supplementary Fig. 6C). We analyzed the extent of gallium oxidation over different cycles of heating and cooling. Although the oxide layer thickness increases during the initial cycles, it stabilizes after 5 cycles and remains constant until 15 cycles (Supplementary Fig. 7A). More details are given in Supplementary Note 2 observation and analyzes of gallium oxide. The second issue is whether the phase change will affect the shape and adhesion of the Ga stamp, considering the repeated phase transitions in the transfer printing process. Supplementary Fig. 8 presents precise photographs capturing 25 cycles of heating and cooling for the gallium stamp. To quantitatively validate the shape consistency, we overlaid the outlines of the liquid gallium droplet from the first and last heating cycles (Supplementary Fig. 9A). We conducted a correlation

coefficient analysis using a Matlab program (Matlab Version: R2019a, the relevant codes are given in Supplementary Data 4), demonstrating a high correlation coefficient of 0.9995. Additionally, we performed pixel-wise correlation analyses by a Matlab program on 50 images of solid and liquid gallium stamps over 25 cycles, generating heatmaps that show consistency coefficients above 0.9945 for all images (Supplementary Fig. 9B). These results confirm the excellent shape consistency through multiple phase transitions. To further analyze the impact of increasing oxide layer thickness on the adhesion strength of gallium metal stamps, adhesion tests were conducted after 25 cycles of heating and cooling (Supplementary Fig. 7B). The experimental results show that the oxide layer has minimal impact on the high adhesion of solid gallium and the low adhesion of liquid gallium. Consequently, the adhesion strength of the gallium metal stamps remains nearly unchanged even after multiple heating and cooling cycles. Such thickness change of oxide nanolayer does not significantly affect the adhesive properties, thus ensuring the reusability of the gallium stamps. The third issue is the reuse of gallium stamps. Although the shape of the Ga droplet can remain consistent over cycles of heating and cooling, for the integrity of droplet shape and to prevent potential issues associated with storage, after each printing, we completely melt the localized solidified stamp, withdraw it into the syringe, and then extrude a well-shaped gallium droplet again, as mentioned in Materials and Methods (Supplementary Fig. 10A, Supplementary Movie 2). During the withdrawal of a small liquid gallium stamp into a syringe already containing a significant volume of liquid gallium (Supplementary Fig. 10A), mechanical forces continuously fracture the nano-Ga oxide layer on its surface. The disrupted oxide layer integrates into the larger liquid metal volume without compromising the reusability of gallium stamps[62]. Upon re-extrusion, exposure to air prompts the immediate formation of a new oxide layer on the gallium stamp's surface, preserving its rounded shape and high surface tension. To verify the adhesive properties of the recycled LM stamps, we conducted adhesion tests over 10 cycles (Supplementary Fig. 10B), which demonstrates that the adhesion of the recycled LM stamps remained nearly unchanged after 10 cycles of withdrawal and re-extrusion.

## Adhesion of the LM stamp during transfer printing

To prevent cracks and wrinkles from occurring in the inorganic thin films, we conducted experiments and numerical analysis to study the adhesion of the LM stamp. During the pickup step, a preload is usually applied to provide high adhesion and ensure the conformal contact of the stamp to the inks, which may cause damage to the inks. The LM stamp holds great potential to eliminate this undesired preload effect due to its intrinsic liquid nature. The experimental optical photos record a liquid Ga droplet approaching a silicon film (5 mm × 5 mm × 100 μm) on a glass substrate with a compressive strain of up to 10% (Fig. 2A, the first row), and these experimental results are compared with those of an unstructured, flat PDMS stamp (Fig. 2A, the second row). Figure 2B shows that the preload of the PDMS stamp applied to a silicon film is up to an order of magnitude greater than that of the liquid Ga stamp under the same applied compressive strain, which indicates that the fluidity of the liquid Ga stamp can protect the inks from being destroyed.

To analytically obtain the relationship between the preload and the compressive strain, a theoretical model is established for the contact of the liquid droplet to the thin film. The liquid droplet to be investigated is illustrated in Fig. 2C. A cylinder with radius $x_1$ is assumed to be in the approaching process towards a silicon thin film through a liquid Ga droplet. The contact angles of the droplet with the cylinder and the silicon film are $\theta_1$ and $\theta_2$, respectively. The periphery of the liquid Ga is pinned at the edge of the cylinder. Both the liquid volume and the contact angle $\theta_2$ are conserved as the increase of compression, while the contact angle $\theta_1$ and the radius of the wet area on the thin film $x_2$ may change to accommodate the compression. The contact force $F$

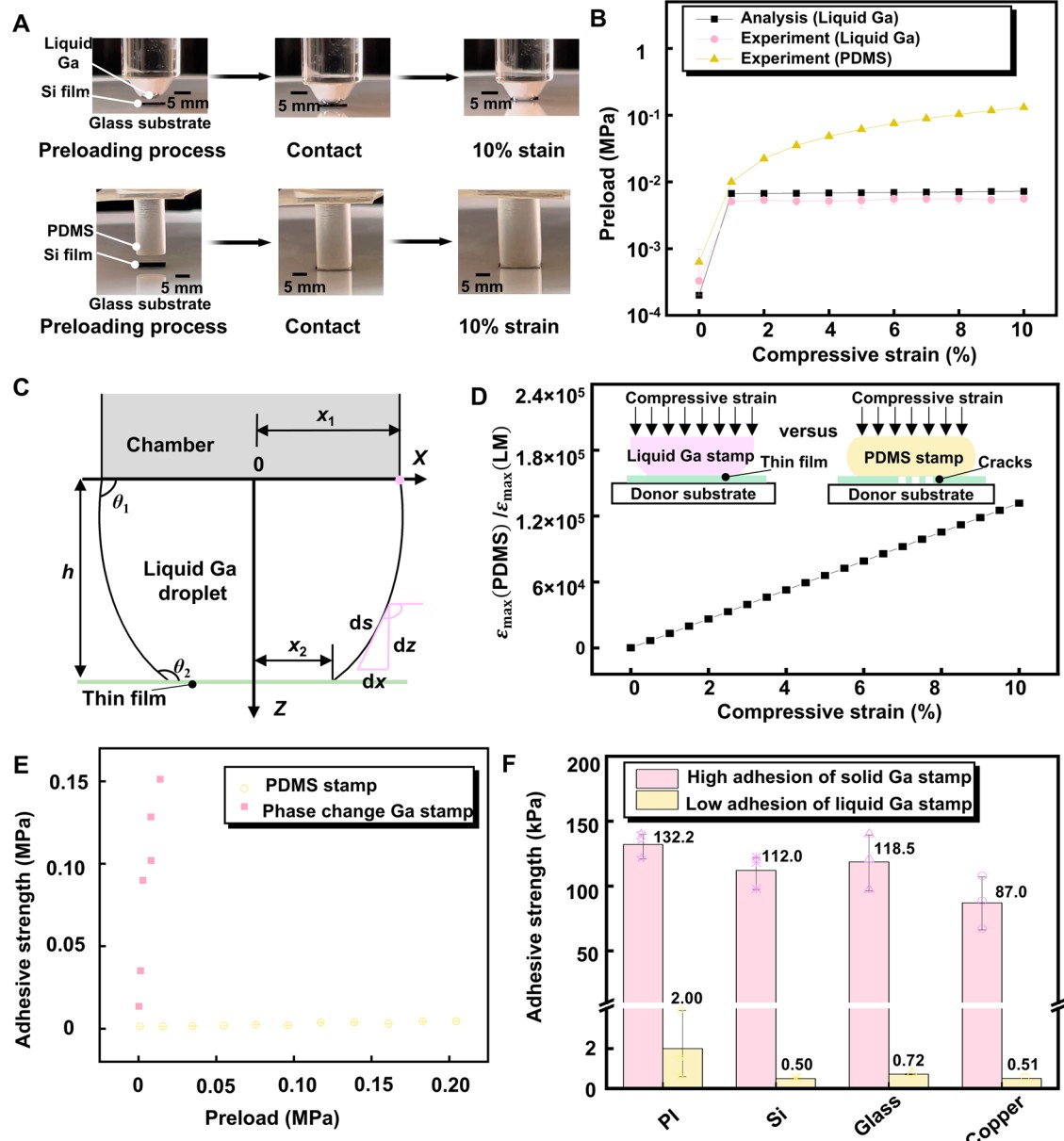

**Fig. 2 | Characterization of the phase change gallium (Ga) stamp. A** Optical images illustrating the approach of the liquid Ga stamp and polydimethylsiloxane (PDMS) stamp to the silicon films, which includes three periods, first approach, then contact until the strain up to 10%. **B** Preload versus compressive strain. **C** Illustration of the geometry of the liquid Ga stamp between the film and the cylindrical chamber, with the cylinder radius, the contact angles $\theta_1$ and $\theta_2$, the radius of the wet area on the thin film $x_2$, the cylinder height $h$. **D** The maximum shear strain ratio of to in silicon films versus compressive strain of stamps. Insets: illustrations of cross-sectional schematics in the approaching process of stamps. **E** Comparison of adhesive strength as a function of preload between the PDMS stamp and the phase change Ga stamp. **F** The adhesion strength of the phase change Ga stamp in high and low adhesion states on different materials. The standard deviation is based on 3 repeated experiments. Source data are provided as a Source Data file.

acting on the thin film consists of two components. One component is induced by the pressure difference across the liquid–gas interface, while the other arises from the axial component of liquid surface tension acting along the liquid perimeter on the edge of the contact area. The control force $F$ can then be expressed as[40,63] $F = -\pi x_2^2 \cdot \triangle P + 2\pi x_2 \cdot \gamma \sin(\pi - \theta_2)$, where $\gamma$ is the liquid Ga surface tension and $\Delta P$ is the pressure difference inside and outside the droplet, which is related to the local liquid Ga profile by the Young–Laplace equation $\triangle P = \gamma(1/R_1 + 1/R_2)$ with $R_1$ and $R_2$ as the two local principal curvatures of the liquid profile. The calculation of preload requires the liquid Ga profile for given liquid volume $V$ and contact angles $\theta_1$ and $\theta_2$ with details given in in Supplementary Note 1: calculations of adhesive force and shear strain. Considering a quasi-

static loading of the liquid Ga droplet onto the thin film, the preload stress can be obtained from the contact force divided by the contact area. The analytical predictions, validated by experimental measurements with the compressive strain up to 10%, are shown in black in Fig. 2B.

An existing study[49] reveals that thin films can rupture due to shear stresses when vertically compressing soft stamps to pick up silicon films from their donor substrates. The approaching processes involving a liquid Ga droplet and a PDMS stamp are illustrated by cross-sectional schematics in the insets of Fig. 2D, and there is no horizontal restraint on the interfaces between the thin film and the donor substrate. Firstly, the stamp makes gentle contact with the thin film that is sitting on the donor substrate. When the external compressive strain is

applied to the stamps to establish intimate contact with the thin films, a high shear strain resulting from Poisson's effect can eventually cause thin films to crack. Therefore, it is necessary to calculate the maximum shear strain of the thin film under compression at the stamp-film interface generated from the liquid Ga stamp and the PDMS stamp. Because the shear stress applied from a liquid droplet equals the horizontal component of surface tension in the plane (Fig. 2C), the maximum strains of the thin film from the liquid Ga stamp and PDMS stamp can be calculated as[49] $\varepsilon_{max}(LM) = \gamma \cos(\pi - \theta_2)/[(2\mu_m - \lambda_m)h_m]$ and $\varepsilon_{max}(PDMS) = Ea^2\varepsilon/\left[(2\mu_m - \lambda_m)h_{pdms}h_m\right]$, respectively, where $h_m$ and $h_{pdms}$ are the thickness of the silicon film and the PDMS stamp, respectively, $\mu_m$ and $\lambda_m$ are the Lame contents of the silicon film which are two material dependent quantities calculated in terms of Young's modulus ($E_m$) and Poisson's ratio ($\nu_m$) of Si (i.e., $\mu_m = E_m/\left[2(1+\nu_m)\right]$ and $\lambda_m = \nu_m E_m/\left[(1 - 2\nu_m)(1 + \nu_m)\right]$), $a$ is the half length of the silicon film, and $E$ and $\varepsilon$ are the Young's modulus and compressive strain on PDMS stamp, respectively. As illustrated in Supplementary Fig. 11, the maximum shear strain in silicon film from the PDMS stamp $\varepsilon_{max}(PDMS)$ obviously increases linearly with the compressive strain, while that from the liquid Ga stamp $\varepsilon_{max}(LM)$ is almost unchanged. The material constants and geometric dimensions in the above calculations are given in Supplementary Note 1: calculations of adhesive force and shear strain. The ratio of $\varepsilon_{max}(PDMS)$ to $\varepsilon_{max}(LM)$ in Fig. 2D linearly increases with the increase of compressive strain. The liquid stamp can reduce the shear strain in silicon film significantly. For example, under a compressive strain of 5%, the shear strain in silicon film from the LM stamp is four orders of magnitude lower than that from the PDMS stamp, which avoids any damage during transfer printing.

Even under a gentle preload, the metal Ga stamp can exhibit a high adhesion due to its phase change. The adhesion strength $F_c$ during the interface separation is positively correlated with the contact area $A$, system stiffness $K$, and interfacial energy release rate $G_c$, as $F_c \sim \sqrt{G_c A K}$[50,64]. During the transition of liquid metal to solid Ga, the substantial increase in system stiffness is primarily attributed to the transition of gallium's Young's modulus from the order of Pa to the order of GPa. Consequently, there is a significant enhancement in the adhesion strength at the interface. Before picking up the micro/nano-thick membranes, the liquid Ga solidifies into a solid state with high adhesion properties. Typical force-displacement curves are shown in Supplementary Fig. 12, in which the maximum preload of the PDMS stamp and liquid Ga stamp are 0.19 MPa and 0.0053 MPa, respectively. The liquid Ga stamp with the outer heating stimulus and the PDMS stamp separately approached the silicon substrates up to 15% strain at a speed of 30 μm/s, followed by removing the outer stimulus and cooling the LM stamp to room temperature (-20 °C) with the preload remaining unchanged, while the PDMS stamp had no phase transition. The retraction of the stamps at a speed of 300 μm/s gives the pull-off force, which yields the adhesive strength under the high adhesion state, which is 0.15 MPa for solid Ga stamp and 0.01 MPa for PDMS stamp. Furthermore, vertical pull tests were carried out under various preloads to compare the adhesion of the stamp/silicon interface between the metal Ga stamp and PDMS stamp, as illustrated in Fig. 2E, of which the top pink point of the Ga stamp and rightmost yellow point of PDMS stamp are the preload and adhesion results in Supplementary Fig. 12. The maximum adhesive strength of the solid Ga stamp is two orders of magnitude larger than that of the PDMS stamp, while the corresponding preload of the liquid Ga stamp is two orders of magnitude smaller than that of the PDMS stamp. Compared to the traditional PDMS stamp, the metal Ga stamp combines the advantages of low preload and high adhesive strength. Demonstration of the pickup and printing of a Si film (5 mm × 5 mm × 100 μm) based on the liquid metal stamp indicates the effectiveness of the proposed PLMT strategy (Supplementary Fig. 13). Simultaneously, we conducted tests to evaluate the adhesion strengths of the metal Ga stamp to various materials, including polyimide (PI), silicon, glass, and copper, in both high

adhesion and low adhesion states with the adhesion ratios of 66, 224, 164, and 174, respectively (Fig. 2F). These results show that the metal Ga stamp has wide applicability in terms of ink materials. It is worth mentioning that gallium atoms have a continuous solid solution and embrittlement natures with some metals, such as copper, aluminum, platinum, gold, and silver, but copper, platinum, and aluminum have good corrosion resistance by gallium based liquid metals lower at 100 °C[57–59]. In the PLMT strategy, the operating temperature is below 100 °C in a short processing time to protect copper, platinum, and aluminum from corrosion. However, when selecting the PLMT method, it is important to consider the embrittlement nature[57–59] of liquid gallium on certain metals. Thus, it is advisable to avoid using metals as materials for the donor/receiver surfaces. Additionally, directly transferring metal thin films with a gallium stamp is not recommended. it is suggested to have an effective surface protection treatment before using gallium-based liquid metal for prolonged high-temperature treatment[65]. To address this, in the fabrication of the mental sensors in our work, the metal interconnects are encapsulated within a polyimide film (more details in "Fabrication of Flexible Sensor Array" section of the Materials and Methods). Furthermore, to evaluate the impact of surface roughness on the adhesion properties of gallium stamps, we conducted tests using glass substrates with varying roughness levels in Supplementary Fig. 14A, root-mean-square (RMS) roughness ranging from 1.61 nm (smooth, (i)) to 22.78 nm (ii), 118.79 nm (iii), 155.74 nm (iv), and 181.63 nm (v). The results, presented in Supplementary Fig. 14B, indicate that both the high and low adhesion states of gallium on rough glass are lower than those on smooth glass. Adhesion decreases with increasing RMS roughness due to the high-amplitude roughness reducing the real contact area of the liquid Ga on the surface[50,52]. Despite the influence of roughness on adhesion, the high adhesion strength of the phase-change gallium stamps remains relatively high, still exceeding that of dry elastomeric adhesives[52].

## Transfer printing of micro/nano-membranes onto flat and curved surfaces

The PLMT method provides high interfacial adhesion switchability and yield rate in micro/nano-membranes to demonstrate the successful transfer printing of silicon (Si) membranes, microscale resistance sensors, and microscale temperature sensors. Due to the gentle preload and excellent adhesion provided by the liquid metal, silicon membranes with thicknesses of 10 μm, 5 μm, 2 μm, and 600 nm (each measuring 400 μm × 400 μm) were chosen as inks, as they can be easily ruptured by PDMS stamp (Supplementary Fig. 15). In order to establish the robustness and reliability of the PLMT method, an array of silicon membranes was successfully transfer printed onto metal pyramid-shaped receivers (Fig. 3A). This demonstration showcases the method's capacity to print diverse inks onto inclined surfaces. Additionally, the SEM images in the second row of Fig. 3A provide visual insights into the thickness of the silicon membranes. To further illustrate the high-yield performance of the phase change stamp, we selected silicon nanomembranes with a thickness of 600 nm for printing onto both flat and curved surfaces (Figs. 3B and 3C). Figure 3B(i) displays an optical microscopy image of a 2 × 3 array of 600 nm thick silicon membranes on silicon-on-insulator (SOI) with fabrication details provided in "Materials and Methods" before the transfer printing process. The array of the silicon membranes is picked up by the phase change Ga stamp and is integrally printed onto a flat PI film without any disruption (Fig. 3B, ii). Because the nano-thick silicon film is transparent, it allows for a clear view of the parallel metal Au line patterns on the PI film. The magnified images from the camera (Fig. 3B, iii, left) and optical microscope (Fig. 3B, iii, right) exhibit that the 600 nm thick silicon membrane had no damage after transfer printing and the line patterns can be clearly observed through the nanomembrane. To further verify that the PLMT methods do not

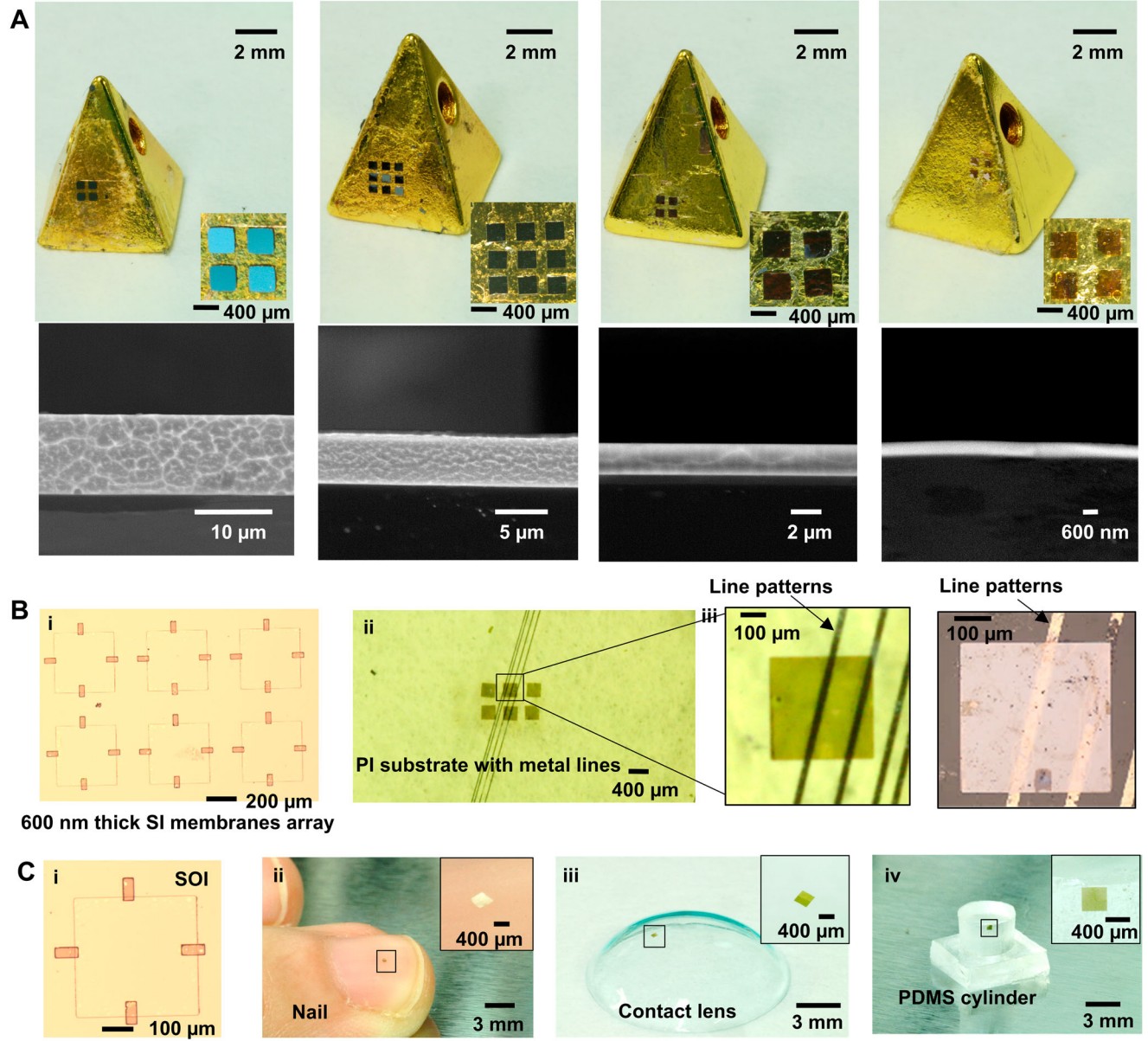

**Fig. 3 | Demonstrations of transfer print Si membranes onto various surfaces.**
**A** Optical images (top row) of the Si membrane arrays (400 μm × 400 μm) transfer printed on a pyramidal inclined plane by the phase change Ga stamp. The scanning electron microscope (SEM) images (bottom row) of Si membranes with different thicknesses: 10 μm, 5 μm, 2 μm and 600 nm. **B** Transfer printing of 2 by 3 Si nanomembrane array. (i) Preparing Si pellets on an silicon-on-insulator (SOI) wafer.
(ii) Si nanomembrane array on a PI substrate with parallel Au metal line patterns. (iii) The magnified image of the transparency Si nanomembrane by camera (left) and optical microscope (right). Scale bars: 100 μm. **C** Transfer printing of a Si pellet. (i) A Si pellet on an SOI wafer. A Si pellet printed onto unconventional curved substrates such as (ii) a nail, (iii) a contact lens, and (iv) a PDMS cylinder.

damage the 600 nm Si membranes, and consequently do not affect their performance, we refer to recent studies on the crystallization kinetics of gallium under varying conditions[52] and the mechanical stability of gallium-oxide nanofilms encapsulating liquid gallium[54]. Based on these studies, we learned that the silicon nanomembranes are subjected to deformation from the gallium oxide film during pre-load and crystallization. Thus, Atomic Force Microscope (AFM, Cypher ES, Oxford Instruments Technology Co., Ltd) images of the nanoscale Si films were captured before (Fig. 3B, i) and after (Fig. 3B, ii) transfer print, and their surface roughness was analyzed. As shown in Supplementary Figs. 16A and 16B, there were no significant differences in surface roughness between before (RMS roughness lower than 0.96 nm) and after transfer printing (RMS roughness lower than

0.99 nm). In Supplementary Fig. 16C, the resistivity results of a Si nanomembrane are ~7.5 ohm·cm and ~7.8 ohm·cm before and after transfer printing processes, respectively. This indicates that the transfer process does not impact the surface morphology and per-formance of Si nanomembranes. To understand the underlying mechanisms, we utilized molecular dynamics (MD) simulations at the nanoscale to reflect the corresponding microscopic mechanical behavior in the dynamic contact process between the Ga-based stamp and the silicon nanomembrane. The utilized forcefield in MD simula-tions can accurately describe the interactions among atoms, and the collective behavior of atoms can represent the strained condition of the material system based on the principle of statistical mechanics. Therefore, we conducted the simulations to investigate the gallium

oxide film changes under pressure and crystallization kinetics during the liquid-to-solid transition of gallium droplets in contact with silicon nanoscale sheets during the PLMT process, and their impact on the performance of nanoscale sheets. In Supplementary Fig. 17A, the simulations showed the local atomic strain distribution after compressing the gallium stamp by 40%, revealing that the strain accumulated mainly at the interface, simulation details in Supplementary Note 4: Molecular dynamics (MD) simulation methods. The maximum strain on the gallium oxide layer reached approximately 50% (Supplementary Fig. 17B), while the maximum strain on the silicon wafer remained around 0.1% (Supplementary Fig. 17C, top), far below the silicon's fracture strain of 1%[66]. Considering the volume expansion caused by the crystallization of gallium metal when it has already been compressed 40% from liquid to solid, the maximum strain on the silicon wafer increased to around 0.5% (Supplementary Fig. 17C, bottom), which is still below the fracture strain of 1%. The MD simulation results indicate that although the silicon sheet is subjected to deformation from the gallium oxide film during preload and volume expansion during crystallization, the induced strain on the nanoscale silicon sheet is minimal and does not affect its performance. Considering the pattern consistency before (Fig. 3b(i)) and after (Fig. 3b(ii)) transfer printing, Si platelets were overlaid before and after the transfer printing process, and the maximum relative displacement and angular changes along with their averages were measured (Supplementary Fig. 18A). In Supplementary Fig. 18B, the positions of the nano-silicon array before transfer are marked with blue dashed outlines, and the positions after transfer are marked with red dashed outlines. We measured the center displacement and rotation angle for each silicon platelet before and after transfer print, and presented the results in bar graphs. Supplementary Fig. 18C shows the center displacement of each platelet along with the deviation percentage relative to a single length of ink, while Supplementary Fig. 18D shows the rotation angle of each platelet. The maximum relative displacement occurred at 53.7 μm on the silicon platelet in the bottom left corner (Supplementary Fig. 18C), while the maximum rotation change took place at 4° on the silicon platelet in the bottom right corner (Supplementary Fig. 18D). To illustrate the consistency in array positions before and after transfer printing, we computed the average displacement and rotation for each silicon platelet. The average displacement across the array was 21.9 μm (-5.4% compared to a single length of ink), and the average rotation was 1.7°. The good pattern consistency observed on the surface of localized molten Ga can be attributed to the thin thickness of the localized molten liquid area to avoid the shift of inks (measurement details in the "Materials and Methods"). In addition, the weak capillary force of liquid gallium will also provide constraint in the relative position of the Si array pattern. After localized melting of the gallium stamp surface by the laser, the relative positions of the Si array almost remain unchanged, as shown in Supplementary Fig. 1D. Such a flat localized liquid area is not as round as a droplet, which is constrained by solid gallium metal and is not easy to flow and deform. The minor displacements and rotations are primarily attributed to manual operation-induced jitter during the printing process and the opaqueness of gallium metal, increasing the difficulty of array alignment as well as the slight curvature of the locally molten stamp when in contact with the flat substrate. To mitigate the dislocation caused by manual operation, we fixed the stamp to a single-axis robotic arm (INSTRON mechanical testing system) to achieve automated pick-up and printing, as shown in Supplementary Fig. 19 and Supplementary Movie 3. During the pick-up process (Supplementary Fig. 19A), the mechanical gripper brought the liquid gallium stamp into contact with the Si platelets (each with dimensions of 200 μm × 200 μm × 2 μm) and solidified to pick up the Si array (optical magnification of Supplementary Fig. 19B). In the print stage (Supplementary Fig. 19C), the localized molten gallium stamp was slowly brought into contact with the PDMS substrate by the mechanical

gripper to complete the transfer printing. Supplementary Fig. 19D shows optical images of the arrays transfer printed by the robotic arm in five times, with the small average relative displacement of the transfer array being 2.35% and the small average rotation angle being 0.715 °. Thus, the use of a mechanical gripper allows precise control of the operation, avoiding disturbances introduced by manual handling. In subsequent printing processes, we will explore multi-degree of freedom equipment-driven operations to minimize errors introduced by manual handling and consider achieving more efficient pattern consistency by improving alignment in future work. For example, we can make a smaller size of the stamp than that of the inks and observe the outline of the inks to achieve alignment, or transfer micro/nano-scale inks to a transparent substrate and observe the alignment from the direction of the substrate. Moreover, the proposed process is not compatible with large-scale printing, in which a larger droplet would increase throughput but bring the localized molten area more round making the pattern dislocate to some extent. When printing on curved surfaces, the localized liquid Ga helps shield the nanomembranes from breaking, allowing them to conform to the curvature of the target surfaces. As a simple demonstration, a 600 nm thick silicon membrane on silicon-on-insulator (SOI) (Fig. 3C, i) can be printed onto curved receivers, such as a nail (Fig. 3C, ii), a contact lens (Fig. 3C, iii), and a PDMS cylinder with a 5 mm diameter (Fig. 3C, iv). These demonstrations confirm that the PLMT strategy is capable of handling ultra-thin, flexible, yet brittle nanofilms on either flat or curved surfaces.

Besides the Si nanomembranes, the microscale resistance sensors (400 μm × 400 μm × 8 μm) were fabricated on a rigid glass to serve as the representative functional inks for transfer printing (Fig. 4A) with the fabrication process given in "Materials and Methods". The microscale resistance sensors were then completely picked up and printed onto the flat surfaces of a smartphone (Fig. 4B) and a starlike acrylic board (Fig. 4C) by phase change Ga stamp. To demonstrate that the PLMT strategy is also applicable for transfer printing of functional film devices onto curvature surfaces, the microscale resistance sensors were sequentially printed to a glass ball with a 5 mm diameter (Fig. 4D), a cone-type acrylic board (Fig. 4E), a PDMS cylinder with 5 mm diameter (Supplementary Fig. 20A, i), and a contact lens (Supplementary Fig. 20A, ii). The resistance of microscale sensors printed onto different receivers using the PLMT method is tested in Fig. 4F. There are no obvious resistance differences before and after the transfer printing, which proves the phase change Ga stamp effectively safeguards the performance of the functional inks. Furthermore, a flexible microscale temperature sensor (2 mm × 2.5 mm × 8 μm) fabricated on a rigid glass (Fig. 4G) was transfer printed onto the soft surfaces, such as a thumb (Fig. 4H) and a PDMS cylinder (Supplementary Fig. 20B, i) by the PLMT method. From the enlarged image in Fig. 4H and S20B, the microscale temperature sensor is well conformal with the complicated skin surface whether the fingers are straight (Fig. 4H) or bent (Supplementary Fig. 20B, iii). Figure 4I shows the measured resistances of the flexible temperature sensor increase as the temperature increases, whose performance has little shift before and after the PLMT method. The slight variations in performance are attributed to measurement errors, such as contact resistance at the probe junctions, from the data acquisition equipment. In summary, the examples provided above illustrate the remarkable capabilities of transfer printing micro/nano-thick inks onto various surfaces using the phase change Ga stamp while preserving their structural and functional integrity.

## Hotplate-directional-induced manipulation of arbitrarily shaped objects

In addition, to transfer print micro/nano-thick membranes, the localized molten Ga stamp also provides a universal gripper that can be easily scaled to maneuver multiscaled objects with 3D arbitrary shapes. Figure 5A shows the hotplate-directional-induced process. The liquid Ga stamp can deform freely and enable intimate contact with 3D

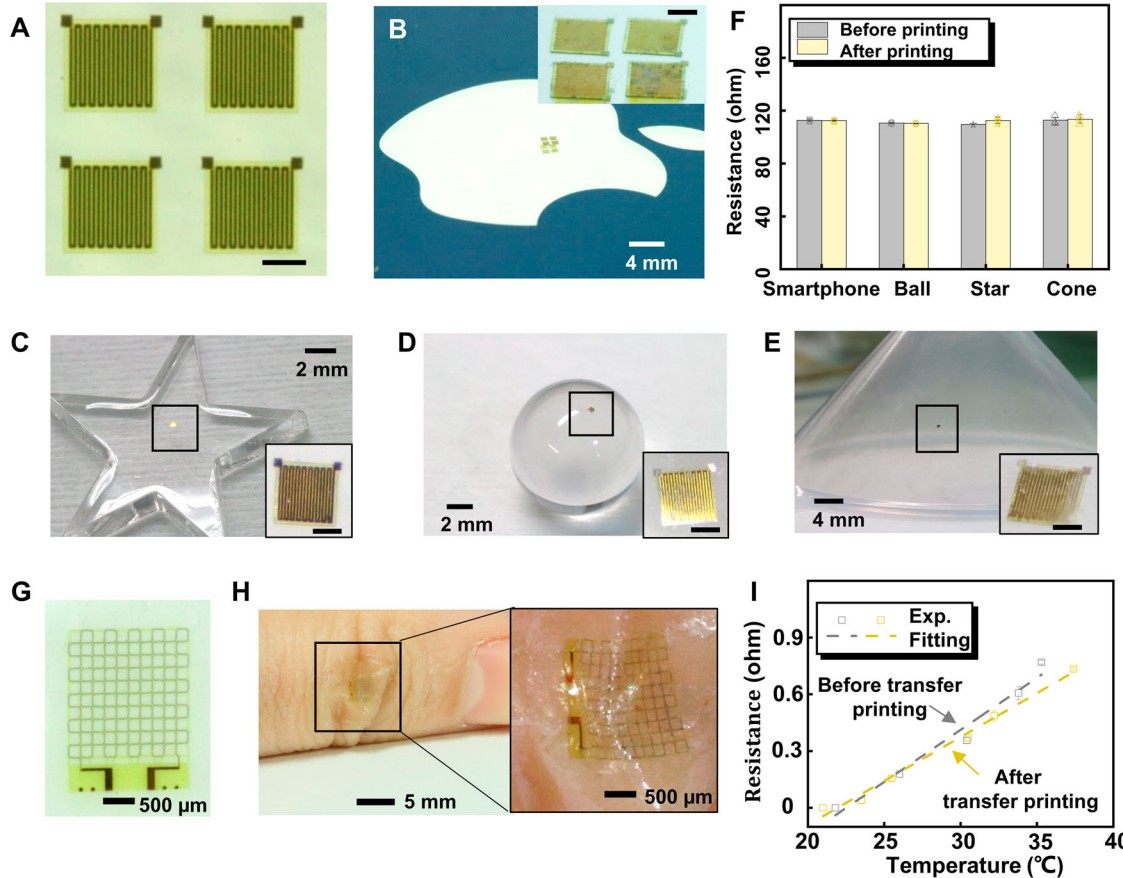

**Fig. 4 | Demonstrations of transfer print functional device onto various surfaces. A** Optical images of the flexible resistance sensor array on a fabricated glass. Printed resistance sensors on (**B**) a smartphone, (**C**) a starlike acrylic board, (**D**) a glass ball, and (**E**) a cone-type acrylic board. Scale bars in (**A**–**E**): 200 μm. **F** The performance of resistance sensors before and after transfer printing onto different surfaces. Optical images of the flexible temperature sensor (**G**) on a fabricated glass and (**H**) printed onto a thumb knuckle. The magnified inset shows the flexibility of the sensor. **I** The measured resistance of the temperature sensor as a function of temperature before and after transfer printing on the thumb knuckle. The standard deviation is based on 3 repeated experiments. Source data are provided as a Source Data file.

arbitrarily shaped objects (Fig. 5A, i). Then the liquid Ga stamp is solidified and becomes stiff with the elastic modulus of 9.3 GPa to fix the temporary shape and lock the target objects after the phase transition (Fig. 5A, ii), which offers a large grip force due to the interlocking effects for gripping and manipulation. Upon hotplate-induced directional heating of solid Ga from the ink side, the localized Ga stamp recovers to a liquid state with a weak adhesion at the contact area with the ink, which enables the release of target objects (Fig. 5A, iii). The ability to pick up and release 3D objects is demonstrated by manipulating a 5 mm diameter steel ball using the phase change Ga stamp in Fig. 5B. The liquid Ga in a chamber with a diameter of 4 mm is used to demonstrate the viability of the universal metal Ga stamp in gripping macroscale objects. Supplementary Fig. 21 presents photographs and thermograms of the gallium metal stamp throughout a complete cycle of picking and printing a macroscale Si plate (5 mm × 5 mm × 100 μm). The initial temperature of solid Ga was 20.1 °C (Supplementary Fig. 21A) before heating. It subsequently rose to 86.2 °C during a 10-minute process with a hotplate (Supplementary Fig. 21B), transitioning into liquid gallium (Supplementary Fig. 21C). Following this, the liquid gallium metal contacted a macroscale silicon plate (Supplementary Fig. 21D) and gradually cooled to 20.3 °C over 9 minutes at room temperature (Supplementary Fig. 21E). Throughout the cooling phase, the solidified gallium metal exhibited strong adhesion, facilitating the picking of the silicon plate from the paper substrate (Supplementary Fig. 21F). Before release, the gallium stamp, in contact with the silicon plate area, was hotplate-induced directional heated for 4 minutes,

localized melting at around 63.9 °C (Supplementary Fig. 21G). This was followed by the printing of the silicon plate onto a glass substrate (Supplementary Fig. 21H). Figure 5C shows the snapshots of gripping macroscopic objects with varying sizes and arbitrary shapes, including a Z-shaped metal model (Fig. 5C, i), J-shaped metal model (Fig. 5C, ii), U-shaped metal model (Fig. 5C, iii), a 20 g weight (Fig. 5C, iv), a bolt (Fig. 5C, v), a 5 mm diameter steel ball (Fig. 5C, vi), a diamond-shaped glass model (Fig. 5C, vii), and a key (Fig. 5C, viii). Supplementary Fig. 22 realizes the process of transfer printing a large area Si wafer with a 10 cm diameter through a 5 mm diameter gallium stamp, which demonstrates that small solid gallium stamps have such high adhesion strength to manipulate large objects. It is worth noting that the large ink is affected by gravity much more than the capillary force of liquid Ga, but the ink can be in contact with the target substrate during printing to effectively prevent the effect of gravity.

To demonstrate the massive manipulation capabilities of the PLMT method, a 3 by 3 array of steel spheres with a diameter of 500 μm is initially arranged on a steel donor (Supplementary Fig. 23A), and then picked up by liquid Ga and subsequently fixed by the solidified Ga (Supplementary Fig. 23B). As further demonstrated in Fig. 5D, this array of steel spheres is successfully printed onto unconventional substrates, including a kumquat (Fig. 5D, i), an Osmanthus Fragrans leaf (Fig. 5D, ii), a person's forearm (Fig. 5D, iii), and a mouse (Fig. 5D, iv). Since these substrates are non-sticky, we printed balls on different receivers with an interlayer adhesive (50 μm thick VHB double-sided tape) to avoid the ball rolling on the

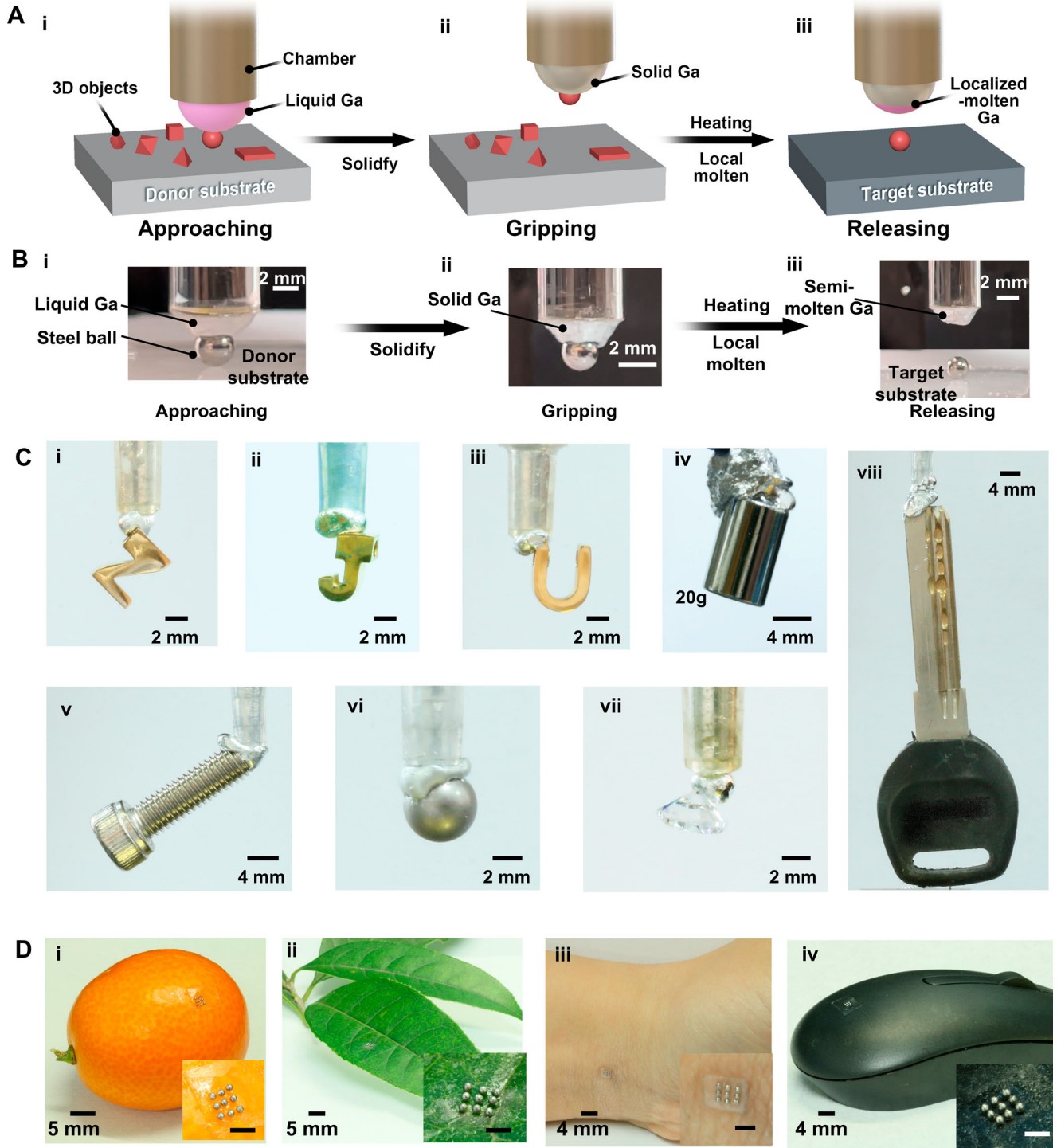

**Fig. 5 | Manipulation of 3D arbitrarily shaped objects. A** Schematic illustration and (**B**) optical images of the pickup and printing process of 3D objects. **C** The universal metal Ga gripper holds various objects such as (i) a Z-shaped metal model, (ii) a J-shaped metal model, (iii) a U-shaped metal model, (iv) a 20 g weight, (v) a bolt, (vi) a ball of 5 mm diameter, (vii) a diamond-shaped model, and (viii) a key.

**D** Printing of a 3 by 3 array of steel spheres with a diameter of 500 µm onto unconventional substrates such as (i) a kumquat, (ii) an osmanthus fragrant leaf, (iii) a forearm of a person, and (iv) a mouse. Insets show the magnifications of the printed arrays of steel spheres. Scale bars of insets: 2 mm.

substrate. These demonstrations illustrate the reliable printing achieved by the PLMT method without disrupting the original order to some extent, owing to the combined effects of the high adhesion ratio of phase change Ga and the point contact area on the surface of the spheres array in favor of the conformal of the array on non-planar surfaces. But it's worth noting that there are still some uncertainties

in the printing process. In addition to some of the effects of manual manipulation and opaque Ga stamps, the point contact force between the steel ball array and the non-planar surface and the capillary force of Ga droplets on the ball is weak which makes it more difficult for the steel ball array to achieve pattern consistency, so there may be a little position deviation.

## Discussion

To conclude, this study presents a robust, damage-free, yet simply operable transfer printing technique to manipulate micro/nano-membranes and arbitrarily shaped objects, which enables strong adhesion for a reliable pickup of inks and weak adhesion for successful printing inks onto arbitrary surfaces. These characteristics are realized by localized molten metal Ga through precision-induced external thermal stimulus. While localized heating for elastomer stamp transfer is an already well-established technique, the proposed localized molten stamp features a gentle contact force and exceptional conformal adaptability and yet improves operation reliability when compared to the fully liquefied Ga stamp. Enabled by the fluid behavior of the liquid Ga, the stamp can deform freely and have intimate contact with inks without any damage in the contacting process. During the pickup process, the solidified Ga stamp demonstrates high adhesive strength as it hardens and securely adheres to the Si micro/nano-membranes and functional micro membrane sensors. This robust adhesion also allows it to effectively interlock with macroscopic arbitrarily shaped objects. In contrast to the destructive effect of hard contact from traditional rubber stamps, the inorganic materials maintain their structural integrity and functionality by the proposed PLMT strategy based on the phase change Ga stamp. The innovative PLMT strategy opens up a simple, efficient, yet robust route toward the integration of ultrathin and delicate functional micro/nano-scale objects in a damage-free fashion.

## Methods

### Ethics statement

This has been approved by Science and Technology Ethics Committee, College of Biomedical Engineering and Instrument Science, Zhejiang University, with protocol number [2023]33 and the person in the demonstration provided informed consent to participate in the transfer demonstration (Figs. 4 and 5).

### Preload force and adhesion test

The preload force and adhesion on the silicon films of the metal Ga stamp and the PDMS stamp were measured by pull tests using an INSTRON mechanical testing system (Model 5944). The approach of the stamp to a silicon film at a speed of 30 μm/s recorded the preload force, and the retraction of the stamp at a speed of 300 μm/s recorded the adhesion.

### Preparation of LM droplet

To prepare the LM droplet, liquid Ga was injected into a syringe (1 mL, 4 mm diameter gauge), and the LM droplet was extruded through the syringe (or needle with 0.52 mm diameter gauge) by manual operation. The volume of the droplets could be determined by the diameter of the syringe/needle. Generally, the diameter of the syringe (or needle) constrains the maximum extrusion amount of the droplet, liquid metal droplets may fall from them when the volume becomes too large. Typically, a syringe with a 4 mm diameter and a needle with a 0.82 mm diameter can extrude droplets with heights of approximately 5 mm and 2.46 mm, respectively. The extruded LM droplet could be used in the PLMT process.

### Laser-transient-induced heating process

As shown in Supplementary Fig. 24, a laser heating apparatus is composed of a near-infrared laser system (FC-W-808 nm-13 W, Changchun New Industries Optoelectronics Tech. Co., Ltd, China), an optical system for in situ monitoring and a translational stage for the alignment between the ink and the target. The translational stage system is composed of an automated positioning system (Linear stage PSA200, Zolix Inc, China) and a manual triaxial linear stage (EB-050-M-N/F, Everbeing Int'l Corp) to control the movement and alignment of the stamp with the laser spot (1 - 2 mm diameter), in which the placement accuracy for the in-plane movements (the X and Y stages) is 5 μm and 0.9 μm, respectively. The accurate position of the laser spot is aligned with the solid Ga surface where Si platelets are located, as illustrated in Supplementary Fig. 1B, which can be targeted from the optical system of the display and adjusted through the translational stage system. The solid Ga stamp was placed on the rotation/tilt stage of the automated positioning system to be laser-heated with a lower power density of 0.82 W/mm². By measuring the difference in thickness of the solid part of the gallium stamp before and after laser heating, the thickness of the localized liquefied gallium metal is ~90 μm. The laser ON time and power can be input and set through the software interface of the laser system, allowing for automatic control by the laser system.

### Phase Transitions of Metal Ga Stamp

During the picking process, the solid metal gallium (99.99%, Shanghai Aladdin Biochemical Technology Co., Ltd.), with a melting point of 29.76 °C, can be melted above 30 °C and fully melted at 86.2 °C (Supplementary Fig. 21) through a non-contact temperature field provided by a hotplate (8786D, Delixi Electric Co., Ltd., China). During the micro/nano films printing process, the solid metal gallium can be localized melted above 30 °C (Supplementary Fig. 1–C) by a commercially near-infrared laser system (FC-W-808 nm-13 W, Changchun New Industries Optoelectronics Tech. Co., Ltd, China). When turning to print macroscale or 3D objects with large contact areas, due to the limitation of the operating space of the laser equipment, we chose a hotplate to provide a direction-induced temperature field for heating from the ink side. The surface of solid Ga is also localized melting first at the temperature around 63.9 °C (Supplementary Fig. 21G and Supplementary Movie 4), and the localized melted gallium stamp is conducive to efficient retraction and avoids the risk of dripping itself. The liquid gallium stamp is then fully heated and withdrawn by a syringe for reuse.

### Fabrication of ultrathin Si nanomembrane array

To prepare the silicon nanomembrane array, an SOI wafer (Suzhou research material micro nano technology Co., Ltd, China) with 600 nm thick single crystalline Si on top was cleaned and spin-coated with AZ 5214 photoresist at 2000 rpm for 40 s and then baked on a hotplate at 110 °C for 60 s. The geometry of the Si square pellet (400 μm by 400 μm) of the 69 by 69 array was defined by ultraviolet photolithography with a dose of 200 mJ/cm², followed by development for 50 s by an NMD 2.38% developer. Thereafter, the Si array was patterned by inductively coupled plasma (ICP) etching for 60 s under a sulfur hexafluoride environment [gas flow, 50 standard cubic centimeters per minute (sccm)]. Then, the remaining photoresist and exposed $SiO_2$ were removed by immersing in acetone for 5 mins and hydrofluoric acid (49%) for 2 mins, respectively. To avoid the floating away of Si pellets after the full removal of the buried $SiO_2$ layer, the photoresist anchors for the Si pellet array were formed by photolithography. The Si pellet array was finally separated from the SOI wafer by immersing it in concentrated hydrofluoric acid (49%) for 180 minutes, following complete undercut etching of the $SiO_2$ buffer layer. The fabrication process of the Si platelets with 10 μm, 5 μm, and 2 μm thickness follows similar procedures.

The fabricated ultrathin Si platelets can be printed onto different receivers without an interlayer adhesive due to the van der Waals force between thin films and receivers[36,42,67]. For two flat surfaces, the maximum attractive van der Waals force per unit area[68] is $P_{vdW} = A_H / 6\pi D_0^3$, where $A_H$ is the Hamaker constant, and $D_0$ is the surface gap distance. The Hamaker constants[69,70] for thin films with common receivers PI, Si, glass, and copper are $28.3 \times 10^{-19}$ J, $23 \times 10^{-19}$ J, $8.53 \times 10^{-19}$ J, and $23.4 \times 10^{-19}$ J, respectively, with a surface gap distance generally less than 10 nm[71]. Therefore, the calculated adhesive strength between the thin film and the four receivers is greater than 15.01 kPa, 12.20 kPa, 4.53 kPa, and 12.41 kPa, respectively, which is approximately

1-2 orders of magnitude stronger than the weak adhesive force of liquid metal (Supplementary Fig. 25).

## Fabrication of flexible sensor array

To prepare the resistance sensor array, the polyimide precursor (ZKPI-305IIE, POME) was spin-coated on a glass wafer at 2000 rpm for 60 s and cured on a hotplate at 80 °C for 60 min, 110 °C for 60 min, and 230 °C for 120 min, respectively. The metal layers (Cr/Au, 5 nm/150 nm) were then deposited on the polyimide layer by an e-Beam evaporator and patterned through ultraviolet lithography and the liftoff process. Thereafter, another layer of polyimide as encapsulation was formed, and the whole bilayer polyimide film (8 μm) was patterned by ICP to obtain a resistance sensor array. Considering the potential impact of gallium on metal films, the metal interconnects are encapsulated within a polyimide film to ensure that the gallium stamp does not directly contact the metal interconnects. The sensor array was picked up from a glass substrate using the PLMT method. The fabrication process of the temperature sensor followed similar procedures until ICP etching, and the sensors were released from the glass substrate by immersion in BOE (1:6) then held with a glass slide, and finally picked up using the PLMT method. The resistance data of all sensors was tested by a digital multimeter (Keithley, DMM7510), and the temperature data was tested by an infrared thermometer (UTI89-PRO21, Klein Tools, Inc.). The micro flexible sensors, like ultrathin Si platelets, can be printed onto different receivers without an interlayer adhesive due to the van der Waals force, except for the micro temperature sensor, which is printed onto the thumb joint with 50 μm thick VHB double-sided tape (3 M Company, United States) to prevent the sensor to drop from the finger movements.

## Reporting summary

Further information on research design is available in the Nature Portfolio Reporting Summary linked to this article.

## Data availability

The data that support the conclusion of this study are available within the paper and Supplementary Information. Source data are provided with this paper.

## Code availability

The relevant codes that support the findings of this work are available from the corresponding author and have been uploaded as Supplementary Data 4.

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

## Acknowledgements

The authors J.S. and C.S. acknowledge the supports from the National Natural Science Foundation of China (Grant Nos. U21A20502, 12302214, 12225209, U20A6001, and 12321002), Zhejiang Provincial Natural Sci-ence Foundation of China (Grant No. LQ23A020006), and Shaoxing Yuanju Technology Co., Ltd.

## Author contributions

C.S. and J.S. designed research; C.S., J.J., C.L., and C.C. designed the experiments; C.S. and W.J. analyzed data; C.S. and J.S. wrote the paper. All authors have given approval to the final version of the manuscript.

## Competing interests

The authors declare no competing interests.

 
