## [Transparent Peer Review file · Nature Communications]

Precision-Induced Localized Molten Liquid Metal Stamps for Damage-Free Transfer Printing of Ultrathin Membranes and 3D Objects

Corresponding Author: Professor Jizhou Song

Version 0:

Reviewer comments:

Reviewer #1

(Remarks to the Author)

The manuscript demonstrated a precision-induced localized molten technique (PLMT) by the phase transition of metal Ga for damage-free transfer printing of fragile thin films. Based on the technique, microscale resistance sensors can also be completely picked up and printed onto various surfaces. The manuscript is well written, and the concept is worthy of investigation. Therefore, I recommend that the manuscript can be considered for publication after minor revisions. The following comments may help the authors to improve the quality of the manuscript.

1. Because Ga easily forms intermetallic compounds with many metals, the PLMT may not be suitable for transferring metal devices. For instance, in Figure 4I, the performance of resistance sensors has a shift before and after the PLMT. It could be due to the corrosion of Cr by Ga.
2. For the reuse of gallium stamps, 5 cycles of the experiment may be too few.
3. Although the authors have considered the effects of gallium oxide, relevant quantitative analyses are still lacking, such as the specific content of gallium oxide and the adhesion strength changes after many cycles of heating and cooling.
4. In the PLMT, the Ga droplet was extruded by manual operation. Can the Ga stamp be integrated into a robotic arm to achieve a more precise operation?
5. Some format errors. In Figure 1F, it should be "target substrate". In Figure 4C, "2 mm" should be on the same line.

Reviewer #2

(Remarks to the Author)

This study presents a localized molten liquid metal transfer printing technique to achieve printing of micro/nano scale inks/objects without introducing mechanical damages. The comparison between traditional PDMS transfer printing and liquid metal printing proved the advantage of the proposed technology. The gentle contact force and conformal interface have been studied with experiments and simulations. The liquid metal stamp also shows great potential of picking up and transporting arbitrarily shaped objects through the interlocking effects. The work is very interesting overall.

1. Compared with literature [63], this paper has demonstrated the transfer printing of micro/nano scale objects with shows wider applicability. The advantage of localized molten effect is not well characterized. Maybe a comparison between global heating and localized heating can help address this issue.
2. Are there any limitations for the material selection for the donor/receiver surfaces? How does the surface roughness affect this transfer printing method?
3. As shown in Figure S9, the relative dislocation seems to be more severe for the pixels located further away from the center of the array. Is this due to the semi spherical shape of the liquid metal after picking up the array? Does the radius of curvature of the semi-sphere shape affect the precision of the transfer printing for ink arrays?
4. There is a thin oxide layer on the surface of LM. The LM contracts and reflows multiple times. When the LM contracts, what happens to the oxide layer? The oxide would shrink significantly and probably buckles, which could lead to fracture. How did the authors prevent that?

Reviewer #3

(Remarks to the Author)

Reviewer #4

(Remarks to the Author)

This manuscript uses gallium as a stamp material to explore device transfer, leveraging its low melting point for easy phase changes. The core technique involves solidifying gallium droplets after device contact and utilizing Precision-Induced Localized Molten technology to partially melt and place the droplets. However, prior studies have already investigated the pick-and-place operation using gallium, and ongoing research addresses the adhesion properties of gallium in its liquid state. Considering the published literature, this study does not seem to present a novel concept or significant interest to the scientific community. Therefore, the manuscript appears insufficient to merit publication in Nature Communications. The following questions should be addressed for potential revision before submission to another journal:

1. In Figure S9, the maximum relative displacement and angular rotation are only depicted schematically. Please quantify these parameters to substantiate the technique's accuracy and precision.
2. In Figure 3 and supplementary Movie S3, contamination is evident on the device surface post-transfer using the gallium stamp. A related study (Nano Lett. 2018, 18, 4, 2498-2504) discusses contamination changes due to oxide film formation under varying pressure during gallium droplet application on the substrate. It is crucial for this manuscript to quantify the extent of contamination and assess how it impacts device performance, including durability and communication efficiency.
3. According to a referenced study (Adv. Mater. 2016, 28, 5088-5092), pick-and-place applications using gallium have been previously documented. Clarify what distinguishes your method from the referenced work, particularly beyond the size of the transfer device. Also, provide experimental evidence highlighting the differences between localized and global melt transfers of gallium.
4. The novelty claimed in this manuscript is the transfer of wafers at nanoscale heights. Given the existing research on gallium oxide film changes under pressure (Nano Lett. 2018, 18, 4, 2498-2504) and crystallization kinetics dependent on substrate type (Adv. Mater. 2020, 32, 1907453), demonstrate that these factors do not adversely affect the transfer of nanoscale sheets in terms of morphological changes and device performance.
5. The manuscript describes the use of AZ 5214 photoresist on ultra-thin silicon wafers, where both gallium and the photoresist are in contact. Measure and report the adhesive strength between these materials, ensuring that the presence of AZ 5214 does not exacerbate contamination.

Version 1:

Reviewer comments:

Reviewer #1

(Remarks to the Author)

The revised manuscript is now suitable for publication. Most issues have been adequately addressed.

Reviewer #2

(Remarks to the Author)

The authors have reasonably addressed the reviewer comments. The manuscript is recommended for publication.

Reviewer #3

(Remarks to the Author)

Reviewer #4

(Remarks to the Author)

The author has provided a response to my comments; however, I still find the research based on the localized Ga metal transfer via laser heating for the patterning of nm-thick Si layers to be lacking in appeal.

The technique of localized heating for elastomer stamp transfer is already established (PNAS, 2024, 121(5), e2318739121), and there are studies that have demonstrated wafer-scale patterning with larger surface areas than the mm-scale pattern transfer conducted in this manuscript (Adv. Mater. Technol. 2020, 5, 2000549).

Therefore, I do not support this study is suitable for publication in Nature Communications.

- There are already studies that have successfully transferred small chips as small as 4 μm in size (Nature, 2023, 614, 81–87). Please explain the minimum chip size that can be transferred using the technique presented in this manuscript. Additionally, compared to other studies that have transferred micro-scale chips, please elaborate on the technical advantages of your approach.

- There are also studies that have performed large-area wafer-scale patterning (Adv. Mater. Technol. 2020, 5, 2000549). It would be important to provide an explanation of the maximum area that can be patterned using your technique.

Reviewers' Comments and Our Responses

Reviewer #1:

The manuscript demonstrated a precision-induced localized molten technique (PLMT) by the phase transition of metal Ga for damage-free transfer printing of fragile thin films. Based on the technique, microscale resistance sensors can also be completely picked up and printed onto various surfaces. The manuscript is well written, and the concept is worthy of investigation. Therefore, I recommend that the manuscript can be considered for publication after minor revisions. The following comments may help the authors to improve the quality of the manuscript.

Our response: We thank the reviewer for raising valuable comments. We have addressed the comments accordingly, and we believe that the questions help to improve our manuscript significantly.

1. Because Ga easily forms intermetallic compounds with many metals, the PLMT may not be suitable for transferring metal devices. For instance, in Figure 4I, the performance of resistance sensors has a shift before and after the PLMT. It could be due to the corrosion of Cr by Ga.

Our replies:

Thank you for highlighting this important issue. We have considered the potential impact of gallium on metal thin-film transfer. To address this, in the fabrication of the flexible sensors shown in Figure 4, the metal interconnects are encapsulated within a polyimide film. This encapsulation ensures that the gallium stamp does not come into direct contact with the metal interconnects. Consequently, the performance of the devices remains nearly unchanged before and after the transfer. The slight variations in performance observed in Figure 4I are attributed to measurement errors from the data acquisition equipment in contact resistance at the probe junctions. Detailed information about the device fabrication process has been included in the "Fabrication of Flexible Sensor Array" section of the Materials and Methods to clarify this point.

Our modification to the manuscript:

On Page 20 of the main text, we added "The slight variations in performance are attributed to measurement errors, such as contact resistance at the probe junctions, from the data acquisition equipment.", and we added on Page 26 "Considering the potential impact of gallium on metal films, the metal interconnects are encapsulated within a polyimide film to ensure that the gallium stamp does not directly contact the metal interconnects."

2. For the reuse of gallium stamps, 5 cycles of the experiment may be too few.

Our replies:

We appreciate your concern regarding the number of cycles. To determine if repeated phase changes affect the shape of the Ga stamp during the transfer printing process, we conducted additional experiments over 25 cycles of gallium stamp reuse. The morphology of the liquid metal during solidification and liquefaction over these 25 cycles is shown in Figure S8. Furthermore, we performed pixel-wise correlation analyses using a Matlab program on 50 images of solid and liquid gallium stamps throughout the 25 cycles. The resulting heatmaps demonstrate consistency coefficients above 0.9945 for all images (Figure S9B).

We have included the added information addressing this in the manuscript on page 9. The added experimental results are presented in Figures S8 and S9.

Our modification to the manuscript:

(1) On Page 9 of the main text, we added “The second issue is whether the phase change will affect the shape and adhesion of the Ga stamp, considering the repeated phase transitions in the transfer printing process. Figure S8 presents precise photographs capturing 25 cycles of heating and cooling for the gallium stamp. To quantitatively validate the shape consistency, we overlaid the outlines of the liquid gallium droplet from the first and last heating cycles (Figure S9A). We conducted a correlation coefficient analysis using a Matlab program (Matlab Version: R2019a), demonstrating a high correlation coefficient of 0.9995. Additionally, we performed pixel-wise correlation analyses by a Matlab program on 50 images of solid and liquid gallium stamps over 25 cycles, generating heatmaps that show consistency coefficients above 0.9945 for all images (Figure S9B). These results confirm the excellent shape consistency through multiple phase transitions.”

(2) We modified Figure S8 to show that the shape of the Ga droplet remains consistent over 25 cycles of heating and cooling.

(3) We modified Figures S9A and S9B to show the consistency coefficients for both solid and liquid gallium stamps over 25 cycles of heating and cooling.

Figure S8. The shape of the Ga droplet can remain consistent over 25 cycles of heating and cooling. Scale bars: 4 mm.

Figure S9. (A) The overlapping outlines of the liquid gallium droplet from the first and last heating cycles. (B) The heatmaps of consistency coefficients for both solid and liquid gallium stamps.

3. Although the authors have considered the effects of gallium oxide, relevant quantitative analyses are still lacking, such as the specific content of gallium oxide and the adhesion strength changes after many cycles of heating and cooling.

Our replies:

Thank you for your valuable suggestion. We have conducted additional analyses to address the quantitative aspects of gallium oxide content and the changes in adhesion strength after multiple heating and cooling cycles. We analyzed the extent of gallium oxidation over different cycles of heating and cooling using high-resolution x-ray photoelectron spectroscopy (XPS) in the added Figure S7A. To analyze the impact of increasing oxide layer thickness on the adhesion strength of gallium metal stamps, we conducted adhesion tests after 25 cycles of heating and cooling in the added Figure S7B.

We have included the added information addressing this in the manuscript on page 9 and in Supplementary Materials on Page 4. The added experimental results are presented in Figure S7. And the study by Sun *et al.*, *Sci. Adv.* 10, eadk9460 (2024) was cited in Supplementary Materials and added in Supplementary Reference [7].

Our modification to the manuscript:

(1) On Page 9 of the main text, we added “We analyzed the extent of gallium oxidation over different cycles of heating and cooling. Although the oxide layer thickness increases during the initial cycles, it stabilizes after 5 cycles and remains constant until 15 cycles (Figure S7A). More details are given in *observation and analyses of gallium oxide* in Supplementary Materials.”. And on Page 4 of the Supplementary Materials, we added “(2) Analyses of gallium oxide: We analyzed the extent of gallium oxidation of a 3 mm diameter Ga stamp over different cycles of heating and cooling using high-resolution x-ray photoelectron spectroscopy (XPS, AXIS ULTRA DLD, Kratos Analytical Co. Ltd.)^[7], whose results show five samples (Figures S7A). Previous study demonstrates the oxide nanofilm initially forms with a thickness of one unit cell layer, and then grows in the ambient atmosphere to a maximum thickness of up to 3 nm^[3]. As the oxide layer thickness increases, the pure Ga metal content (centered at approximately 1116.7 eV) in the Ga 2p spectral region shows significant differences. Specifically, the relative amount of pure Ga 2p₃ metal decreases from 28.81% in the original test (Fig. S7A, top left) to 24.29% after 15 cycles of heating and cooling (Fig. S7A, bottom right). Conversely, the amount of Ga 2p₃ oxide and suboxide increases in relative proportion from 71.19% (Fig. S7A, top left) to 75.71% (Fig. S7A, bottom right). This indicates that additional gallium oxide forms in the air as the heating and cooling cycles increase. By approximately the 5th cycle, the gallium oxide growth stabilizes, and the relative proportions of gallium oxide in 5th (75.34%), 9th (75.5%), and 15th cycle (75.71%) remain almost unchanged.”

(2) On Page 9 of the main text, we added “The second issue is whether the phase change will affect the shape and adhesion of the Ga stamp, considering the repeated phase transitions in the transfer printing process.”

(3) On page 10, we added “To further analyze the impact of increasing oxide layer thickness on the adhesion strength of gallium metal stamps, adhesion tests were conducted after 25 cycles of heating and cooling (Figure S7B). The experimental results show that the oxide layer has minimal impact on the high adhesion of solid gallium and the low adhesion of liquid gallium. Consequently, the adhesion strength of the gallium metal stamps remains nearly unchanged even after multiple heating and cooling cycles. Such thickness change of oxide nanolayer does not significantly affect the adhesive properties, thus ensuring the reusability of the gallium stamps.”

(4) We added Figure S7 to show that the extent of gallium oxidation over different cycles of heating and cooling, and the impact of increasing oxide layer thickness on the adhesion strength of gallium metal stamps.

Figure S7. (A) XPS spectra of Ga 2p of the Ga stamp over different cycles of heating and cooling. (B) The high and low adhesion strength of Ga stamp over different cycles of heating and cooling.

4. In the PLMT, the Ga droplet was extruded by manual operation. Can the Ga stamp be integrated into a robotic arm to achieve a more precise operation?

Our replies:

Thank you for your valuable question. Considering the impact of manual operation on transfer precision, we have attempted to use a single-axis robotic arm to achieve transfer printing with improved accuracy. We have supplemented our manuscript with Figure S19 and Movie S3 to demonstrate the transfer process using a robotic arm, and we have calculated the average relative displacement and rotation

angle of a 2×2 array after five transfers. The relevant content has been added to the manuscript. We have included the added information addressing this in the manuscript on page 18.

Our modification to the manuscript:

(1) On Page 18 of the main text, we added “The minor displacements and rotations are primarily attributed to manual operation-induced jitter during the printing process and the opaqueness of gallium metal increasing the difficulty of array alignment as well as the slight curvature of the locally molten stamp when in contact with the flat substrate. To mitigate the dislocation caused by manual operation, we fixed the stamp to a single-axis robotic arm (INSTRON mechanical testing system) to achieve automated pick-up and printing, as shown in Figure S19 and Supplementary movie S3. During the pick-up process (Figure S19A), the mechanical gripper brought the liquid gallium stamp into contact with the Si platelets (each with dimensions of $200 \mu\text{m} \times 200 \mu\text{m} \times 2 \mu\text{m}$) and solidified to pick up the Si array (optical magnification of Figure S19B). In the print stage (Figure S19C), the localized molten gallium stamp was slowly brought into contact with the PDMS substrate by the mechanical gripper to complete the transfer printing. Figure S19D shows optical images of the arrays transfer printed by the robotic arm in five times, with the small average relative displacement of the transfer array being 2.35% and the small average rotation angle being 0.715° . Thus, the use of a mechanical gripper allows precise control of the operation, avoiding disturbances introduced by manual handling. In subsequent printing processes, we will explore multi-degree of freedom equipment-driven operations to minimize errors introduced by manual handling and consider achieving more efficient pattern consistency by improving alignment in future work.”

(2) We added Figure S19 and Supplementary movie S3 to demonstrate the transfer process using a single-axis robotic arm.

Figure S19. (A) The pick-up process of the PLMT method conducted by a single-axis robotic arm. (B) A 2×2 square silicon platelet array (each with dimensions of $200 \mu\text{m} \times 200 \mu\text{m} \times 2 \mu\text{m}$) picked up by a solid Ga stamp. (C) The print process of the PLMT method conducted by a single-axis robotic arm. (D) Five 2×2 square silicon arrays transfer printed by a single-axis robotic arm on a PDMS substrate.

Supplementary Movie S3

A macroscale Si platelets array transfer printed by localized molten metal gallium through a single-axis robotic arm.

5. Some format errors. In Figure 1F, it should be “target substrate”. In Figure 4C, “2 mm” should be on the same line.

Our replies:

Thank you for pointing out these formatting errors. We have corrected the label in Figure 1F to “target substrate”. Additionally, we have ensured that “2 mm” is now on the same line in Figure 4C. We appreciate your attention to detail.

Our modification to the manuscript:

- (1) We have corrected the label in Figure 1F to “target substrate”.
- (2) We have ensured that “2 mm” is now on the same line in Figure 4C.

Figure 1. Schematic illustration of the pickup and printing process of PLMT.

Figure 4C. Printed resistance sensors on a starlike acrylic board.

Reviewer #2:

This study presents a localized molten liquid metal transfer printing technique to achieve printing of micro/nano scale inks/objects without introducing mechanical damages. The comparison between traditional PDMS transfer printing and liquid metal printing proved the advantage of the proposed technology. The gentle contact force and conformal interface have been studied with experiments and simulations. The liquid metal stamp also shows great potential of picking up and transporting arbitrarily shaped objects through the interlocking effects. The work is very interesting overall.

Our response: We thank the reviewer for raising valuable comments. We have addressed the comments accordingly, and we believe that the questions help to improve our manuscript significantly.

1. Compared with literature [63], this paper has demonstrated the transfer printing of micro/nano scale objects with shows wider applicability. The advantage of localized molten effect is not well characterized. Maybe a comparison between global heating and localized heating can help address this issue.

Our replies:

Thank you for your valuable suggestion. In response, we have conducted additional experiments to validate the efficiency of our PLMT by comparing localized and global molten transfer printing methods. We have included the added information addressing this clarification in the manuscript on page 7. The added experimental results are presented in Figure S3.

Our modification to the manuscript:

(1) On Page 7 of the main text, we added “The localized molten Ga stamp can also help to reduce the dislocation and the pollution of Ga stamp caused by the drop of liquid metal. To validate the efficiency of PLMT, we compared the differences between localized and global molten Ga stamp transfer printing methods. The experimental results are presented in Figure S3. A 2×2 square silicon platelet array (each with dimensions of $200 \mu\text{m} \times 200 \mu\text{m} \times 2 \mu\text{m}$) was initially picked up by a planar solid gallium metal with an infinite radius of curvature (Figure S3A(i)). Subsequently, the array was transferred using both a localized molten gallium stamp, which exhibits a slightly reduced radius of curvature compared to a perfectly planar surface and only undergoes surface flow (Figure S3A(ii)), and a global molten gallium stamp, which forms a single droplet with a radius of curvature decreasing to that of the liquid droplet ($\sim 2 \text{ mm}$) (Figure S3A(iii)). When printing the array onto a PDMS substrate using the localized molten stamp, the array remained highly regular, with minimal gallium

residue around it (Figure S3B, right with a red solid line box). In contrast, the array transferred with the global molten stamp showed significant dislocations and extensive gallium residue (Figure S3B, left with a blue solid line box). Therefore, localized transfer printing achieves higher precision with less contamination compared to global transfer printing. The increased deformation occurs because a droplet with a smaller radius of curvature needs to transition from a curved surface to a flat substrate upon contact. This transition involves greater deformation compared to a localized molten stamp with a larger radius of curvature approaching flatness. Ideally, ensuring contact between a planar stamp and the substrate can maintain transfer precision and prevent dislocation during the printing process.”

(2) We have added Figure S3 to the supplementary materials to demonstrate the differences between localized and global molten transfer printing methods.

Figure S3. Comparison of localized and global molten Ga stamp transfer printing methods. (A) Sequential images illustrating the transfer printing process of a 2×2 square silicon platelet array. (B) Images of the silicon platelet arrays printed on PDMS surfaces by global molten method (left with a blue solid line frame) and localized molten method (right with a red solid line frame).

2. Are there any limitations for the material selection for the donor/receiver surfaces? How does the surface roughness affect this transfer printing method?

Our replies:

(1) Thank you for your insightful questions. It is worth mentioning that gallium atoms have a continuous solid solution and embrittlement nature with some metals, such as copper, aluminum, platinum, gold, and silver. While copper, platinum, and aluminum exhibit good corrosion resistance to gallium-based liquid metals at temperatures below 100 °C. However, when selecting the PLMT method, it is important to consider the embrittlement nature of liquid gallium on certain metals (Can. J. Chem. 98: 787–798 (2020), Resistance Of Materials To Attack By Liquid Metals. ANL--4417, 4419134 (1950), Appl Phys A (2009) 95: 907–915). Therefore, it is advisable to avoid using metals as materials for the donor/receiver surfaces. Additionally, directly transferring thin metal films with a gallium stamp is not recommended. It is suggested to apply an effective surface protection treatment before using gallium-based liquid metal for prolonged high-temperature applications. To address this, in the fabrication of the mental sensors in our work, the metal interconnects are encapsulated within a polyimide film.

We have included the added information addressing this in the manuscript on page 14.

(2) Surface roughness indeed affects transfer printing. Previous studies (Adv. Mater. 2020, 32, 1907453) have shown that the adhesion state of crystalline gallium on rough glass is lower than on smooth glass, but still higher than dry elastomeric adhesives on rough surfaces. We conducted additional experiments to compare the adhesion of gallium metal on smooth and different rough glass surfaces in Figure S14.

We have included the added information addressing this in the manuscript on page 15. The added experimental results are presented in Figure S14. And the study *Adv. Mater. 2020, 32, 1907453* was cited in the manuscript and added in reference [52].

Our modification to the manuscript:

(1) On Page 14 of the main text, we added “However, when selecting the PLMT method, it is important to consider the embrittlement nature^[57-59] of liquid gallium on certain metals. Thus, it is advisable to avoid using metals as materials for the donor/receiver surfaces. Additionally, directly transferring metal thin films with a gallium stamp is not recommended. it is suggested to have an effective surface protection treatment before using gallium-based liquid metal for prolonged high-temperature treatment^[65]. To address this, in the fabrication of the mental sensors in our work, the metal interconnects are encapsulated within a polyimide film (more details in “Fabrication of Flexible Sensor Array” section of the Materials and Methods)”.

(2) On Page 15 of the main text, we added “Furthermore, to evaluate the impact of surface roughness on the adhesion properties of gallium stamps, we conducted tests using glass substrates with varying roughness levels in Figure S14A, root-mean-square (RMS) roughness ranging from 1.61 nm (smooth, (i)) to 22.78 nm (ii), 118.79 nm (iii), 155.74 nm (iv), and 181.63 nm (v). The results, presented in Figure S14B, indicate that both the high and low adhesion states of gallium on rough glass are lower than those on smooth glass. Adhesion decreases with increasing RMS roughness due to the high-amplitude roughness reducing the real contact area of the liquid Ga on the surface^[50,52]. Despite the influence of roughness on adhesion, the high adhesion strength of the phase-change gallium stamps remains relatively high, still exceeding that of dry elastomeric adhesives^[52].”

(2) We added Figure S14A to quantitatively measure the roughness of five different coarseness levels of glass surfaces. Additionally, Figure S14B was added to test adhesion related to varying degrees of roughness.

Figure S14. (A) Atomic force microscope images of smooth and rough glass surfaces with varying roughness. The root-mean-square roughness values are as follows: smooth

glass at 1.61 nm (i), and rough glass surfaces at 22.78 nm (ii), 118.79 nm (iii), 155.74 nm (iv), and 181.63 nm (v). (B) Variation in adhesion strength due to surface roughness. As the roughness of the ink increases, the adhesion strength of the gallium stamp decreases.

3. As shown in Figure S9, the relative dislocation seems to be more severe for the pixels located further away from the center of the array. Is this due to the semi-spherical shape of the liquid metal after picking up the array? Does the radius of curvature of the semi-sphere shape affect the precision of the transfer printing for ink arrays?

Our replies:

Thank you for your insightful question. The observed relative dislocation in updated Figure S18 (original Figure S9) does not originate from the pick-up process but rather primarily occurs during the printing process. We analyzed the effect of radius of curvature on transfer precision in the experimental results of both global and localized transfer printing. During pick-up, the gallium metal is in a solid state, and the surface picking up the platelet array is planar (with an infinite radius of curvature), as shown in added Figure S3A(i). The localized melting part of the gallium stamp causes it to melt only at the ink-contacting surface, resulting in a slightly reduced radius of curvature compared to a perfectly planar surface (Figure S3A(ii)). Upon complete melting, the radius of curvature decreases to that of the liquid droplet (~2 mm), as illustrated in Figure S3A(iii). The transfer printing results indicate that the dislocation for the stamp with a larger radius of curvature (localized molten) (Figure S3B, right) is significantly less than that for the fully molten stamp (Figure S3B, left). That is because a smaller radius of curvature leads to greater deformation when the droplet contacts the flat receiver substrate, resulting in more pronounced array dislocation. Ideally, ensuring planar contact between a flat stamp and the substrate can maintain transfer precision and prevent dislocation during the printing process. We have included the added information addressing this clarification in the manuscript on page 7.

Our modification to the manuscript:

(1) On Page 7 of the main text, we added “The localized molten Ga stamp can also help to reduce the dislocation and the pollution of Ga stamp caused by the drop of liquid metal. To validate the efficiency of PLMT, we compared the differences between localized and global molten Ga stamp transfer printing methods. The experimental results are presented in Figure S3. A 2×2 square silicon platelet array (each with dimensions of $200 \mu\text{m} \times 200 \mu\text{m} \times 2 \mu\text{m}$) was initially picked up by a planar

solid gallium metal with an infinite radius of curvature (Figure S3A(i)). Subsequently, the array was transferred using both a localized molten gallium stamp, which exhibits a slightly reduced radius of curvature compared to a perfectly planar surface and only undergoes surface flow (Figure S3A(ii)), and a global molten gallium stamp, which forms a single droplet with a radius of curvature decreasing to that of the liquid droplet (~2 mm) (Figure S3A(iii)). When printing the array onto a PDMS substrate using the localized molten stamp, the array remained highly regular, with minimal gallium residue around it (Figure S3B, right with a red solid line box). In contrast, the array transferred with the global molten stamp showed significant dislocations and extensive gallium residue (Figure S3B, left with a blue solid line box). Therefore, localized transfer printing achieves higher precision with less contamination compared to global transfer printing. The increased deformation occurs because a droplet with a smaller radius of curvature needs to transition from a curved surface to a flat substrate upon contact. This transition involves greater deformation compared to a localized molten stamp with a larger radius of curvature approaching flatness. Ideally, ensuring planar contact between a flat stamp and the substrate can maintain transfer precision and prevent dislocation during the printing process.”

4. There is a thin oxide layer on the surface of LM. The LM contracts and reflows multiple times. When the LM contracts, what happens to the oxide layer? The oxide would shrink significantly and probably buckles, which could lead to fracture. How did the authors prevent that?

Our replies:

Thank you for your insightful question. To prevent potential issues associated with storage, after each printing, we completely melt the localized solidified stamp, withdraw it into the syringe, and then extrude a well-shaped gallium droplet again. When a small liquid gallium droplet is drawn back into a syringe already containing a substantial amount of liquid gallium (Figure S10A), the oxide layer on its surface fractures and disperses within the bulk liquid metal. This occurs because the mechanical forces during the retraction process continuously break the nano-Ga oxide layer. The disrupted oxide layer integrates into the larger volume of liquid metal and does not affect the reuse of the gallium stamps. Upon re-extrusion, the gallium stamp immediately forms a new oxide layer on the surface when exposed to air, maintaining the stamp’s rounded shape and high surface tension (Mater. Adv., 2021, 2, 7799).

To verify the adhesive properties of the recycled and re-extruded LM stamps, we conducted adhesion tests over 10 cycles. The results demonstrated that the adhesion of the LM stamps remained nearly unchanged after multiple cycles of withdrawal and re-extrusion.

We have included the added information addressing this in the manuscript on page 10. The added experimental results are presented in Figure S10B. And the study *Mater. Adv.*, 2021, 2, 7799–7819 was cited in manuscript and added in reference [62].

Our modification to the manuscript:

(1) On Page 10 of the main text, we added “During withdrawal of a small liquid gallium stamp into a syringe already containing a significant volume of liquid gallium (Figure S10A), mechanical forces continuously fracture the nano-Ga oxide layer on its surface. The disrupted oxide layer integrates into the larger liquid metal volume without compromising the reusability of gallium stamps^[62]. Upon re-extrusion, exposure to air prompts immediate formation of a new oxide layer on the gallium stamp's surface, preserving its rounded shape and high surface tension. To verify the adhesive properties of the recycled LM stamps, we conducted adhesion tests over 10 cycles (Figure S10B), which demonstrates that the adhesion of the recycled LM stamps remained nearly unchanged after 10 cycles of withdrawal and re-extrusion.”

(2) We added Figure S10B to show that the adhesion of the recycled LM stamps remained nearly unchanged after 10 cycles of withdrawal and re-extrusion.

Figure S10. (B) The high and low adhesion strength of Ga stamp over 10 recycle times.

Reviewer #3 (Remarks to the Author):

Reviewer #4 (Remarks to the Author):

This manuscript uses gallium as a stamp material to explore device transfer, leveraging its low melting point for easy phase changes. The core technique involves solidifying gallium droplets after device contact and utilizing Precision-Induced Localized Molten technology to partially melt and place the droplets. However, prior studies have already investigated the pick-and-place operation using gallium, and ongoing research addresses the adhesion properties of gallium in its liquid state. Considering the published literature, this study does not seem to present a novel concept or significant interest to the scientific community. Therefore, the manuscript appears insufficient to merit publication in Nature Communications. The following questions should be addressed for potential revision before submission to another journal:

Our response: We appreciate the constructive feedback provided by the reviewer. We have carefully examined all the feedback and have conducted new investigations to address the issues you raised. The main examples are: (1) we have carried out experiments comparing fully melted droplets and localized molten stamps to demonstrate that localized melting effectively addresses the issues of contamination and high failure rates in thin film device transfer; (2) we have made a detailed explanation of the challenges and innovations of our work in the introduction and have placed enhanced emphasis on the study; (3) we have provided experimental evidence and detailed analysis regarding potential contamination during the transfer process and we indicated that the cleaned devices have the same lifespan as uncontaminated devices; (4) we have also used experimental data and molecular dynamics simulations to explain the effects of Ga_2O_3 changes under force processes during contact and the impact of crystallization on the nano film.

Additionally, we have performed further research according to other reviews' comments to enrich the core content of our work in the manuscript. For example, experimental results found that the thickness of Ga_2O_3 increases and then stabilizes with oxidation. We supplemented our study with extensive adhesion tests under repeated heating, cooling, and recycling conditions. And we analyzed the impact of molten curvature on transfer precision. Furthermore, we utilized a uniaxial mechanical arm for pick-and-place operations to further enhance the transfer precision of thin film arrays.

We thank the reviewer again for raising valuable comments. We have addressed the comments accordingly, and we believe that the questions help to improve our manuscript significantly.

1. In Figure S9, the maximum relative displacement and angular rotation are only

depicted schematically. Please quantify these parameters to substantiate the technique's accuracy and precision.

Our replies:

Thank you for your suggestion. We have updated our analysis to include quantified measurements of the maximum relative displacement and angular rotation. We have measured the center displacement and rotation angle of each silicon platelet before and after the transfer and presented the results in bar graphs. The added Figure S18C shows the center displacement of each platelet along with the deviation percentage relative to a single length of ink, while the added Figure S18D shows the rotation angle of each platelet. We have included the added information addressing this in the manuscript on page 17.

Our modification to the manuscript:

(1) On Page 17 of the main text, we added “In Figure S18B, the positions of the nano-silicon array before transfer are marked with blue dashed outlines, and the positions after transfer are marked with red dashed outlines. We measured the center displacement and rotation angle for each silicon platelet before and after transfer print, and presented the results in bar graphs. Figure S18C shows the center displacement of each platelet along with the deviation percentage relative to a single length of ink, while Figure S18D shows the rotation angle of each platelet. The maximum relative displacement occurred at 53.7 μm on the silicon platelet in the bottom left corner (Figure S18C), while the maximum rotation change took place at 4° on the silicon platelet in the bottom right corner (Figure S18D).”

(2) We added Figure S18C and S18D to quantitatively measurement the maximum relative displacement and angular rotation of each platelet before and after the transfer printing process.

Figure S18. (C) The center displacement of each platelet and the deviation percentage relative to a single length (400 μm) of ink before and after the transfer printing process. (D) The rotation angle of each platelet before and after the transfer printing process.

2. In Figure 3 and supplementary Movie S3, contamination is evident on the device surface post-transfer using the gallium stamp. A related study (Nano Lett. 2018, 18, 4, 2498-2504) discusses contamination changes due to oxide film formation under varying pressure during gallium droplet application on the substrate. It is crucial for this manuscript to quantify the extent of contamination, and assess how it impacts device performance, including durability and communication efficiency.

Our replies:

Thank you for your valuable suggestion. In response, we have conducted additional experiments and included Figure S4 and S5 to address the contamination issue and assess its impact on device performance. We used a gallium stamp with a diameter of approximately 4 mm to contact a $200\ \mu\text{m} \times 600\ \mu\text{m}$ micro-LED, applying different preload pressures to quantify the extent of gallium residue. Minor contamination of gallium metal on printed substrates still exists due to the wrinkle and rupture of the gallium oxide nanofilm under high preload (Nano Lett. 2018, 18, 4, 2498-2504). But the residual Ga can be removed using alcohol (Adv. Funct. Mater. 2021, 31, 2100274). Additionally, we tested the performance of clean, heavily contaminated, and alcohol-cleaned micro-LEDs. After 20 days of voltage-current curve testing, we found that both the clean and cleaned LEDs maintained normal operation, while the micro-LED with gallium residue exhibited normal signals for the first 8 days but started showing issues on the ninth day.

We have included the added information addressing this in the manuscript on page 8. The added experimental results are presented in Figure S4 and S5. And the study *Nano Lett. 2018, 18, 4, 2498-2504* was cited in manuscript and added in reference [54].

Our modification to the manuscript:

(1) On Page 8 of the main text, we added “Although the residual problem is significantly reduced, minor contamination of gallium metal on printed substrates still exists due to the wrinkle and rupture of the gallium oxide nanofilm under high preload^[54]. And the residual Ga can be removed using alcohol^[51]. Therefore, we quantified the extent of gallium metal contamination on devices and evaluated its impact on performance. We selected micro-LEDs (with dimensions of $600\ \mu\text{m} \times 200\ \mu\text{m} \times 2\ \mu\text{m}$) as test subjects and examined the gallium residue on the micro-LEDs under different preload conditions. Figure S4A shows photos of liquid gallium stamps contacting micro-LEDs under five different preload conditions. As the preload increases from 1 mN to 5 mN, the gallium metal is gradually compressed, almost flattening under 5 mN preload. Figure S4B displays the gallium residue under a

microscope, marked with red dotted lines. Initially, we photographed a micro-LED before Ga contact (Figure S4B(i)), where one micro-LED is divided into left and right parts due to high magnification of the microscope. The area ratio of gallium residue increases from 0.1% (1 mN preload) to 0.66% (5 mN preload). Generally, the residue is minimal and mainly located at the micro-LED edges, with a small amount on the gold electrode pads, possibly due to gallium oxide breaking at sharp edges. Finally, we removed the residual Ga with alcohol (Figure S4B(vii)). To test the impact of gallium contamination on device performance, we selected clean, alcohol-cleaned, and micro-LEDs under 5 mN preload micro-LEDs and tested their performance. Figure S5 shows the light-emitting capability of both clean (Figure S5A), alcohol-cleaned (Figure S5B) and contaminated under 5 mN preload (Figure S5C) micro-LEDs in proper function. Additionally, we provided voltage-current curves for both sets of devices during continuous operation. The results indicate that clean and alcohol-cleaned micro-LEDs functioned normally for 20 days (Figure S5D, left and middle), whereas contaminated micro-LEDs signal was abnormal after 8 days (Figure S5D, right). The anomaly is likely because gallium atoms exhibit embrittlement with the gold electrode pads^[57-59]. The cleaned devices have the same lifespan as uncontaminated devices which indicates that the method of cleaning the contamination with alcohol is very effective in maintaining the good working performance of the transfer printed device.”

(2) We added Figure S4 to demonstrating the quantity of gallium residue on device under different preload.

(3) We added Figure S5 to illustrate the impact of gallium contamination on the performance and durability of the micro-LEDs.

Figure S4. (A) photos of liquid gallium stamps (diameter ~ 2 mm) contacting micro-LEDs (with dimensions of $600\ \mu\text{m} \times 200\ \mu\text{m} \times 2\ \mu\text{m}$) under five different preload conditions. (B) Microscopic images of a micro-LED under different conditions, such as clean, alcohol-cleaned, and gallium residual under different preload (marked with red dotted lines).

Figure S5. Light-emitting capability during proper function for micro-LEDs under different conditions: (A) clean, (B) alcohol-cleaned, and (C) contaminated. (D) Voltage-current curves over several days of operation for micro-LEDs under different conditions.

3. According to a referenced study (Adv. Mater. 2016, 28, 5088-5092), pick-and-place applications using gallium have been previously documented. Clarify what distinguishes your method from the referenced work, particularly beyond the size of the transfer device. Also, provide experimental evidence highlighting the differences between localized and global melt transfers of gallium.

Our replies:

(1) Thank you for your insightful question. The referenced study (Adv. Mater. 2016, 28, 5088-5092) indeed demonstrates the feasibility of using the phase change of gallium for strong, reversible, and robust adhesion on various surfaces under different conditions.

Building on this foundational work, we have developed a precise localized transfer printing method to maneuver more challenging devices. These challenges arise due to the difficulty of visual observation, the small mass, and the fragility of the thin films, making the pick-and-place operations significantly more difficult. Our work advances this field by utilizing locally molten gallium stamps to study the transfer techniques specifically for micro- and nanoscale thin films.

Beyond considering the impact of device scale on the difficulty of pick-and-place operations, our research further investigates the effects of the melting extent of the gallium stamp on the precision of the transfer printing process. We found that arrays transferred with the globally molten stamp showed significant dislocations and extensive gallium residue. In contrast, our localized transfer printing method achieves higher precision with less contamination compared to global transfer printing.

We have included the added information addressing this clarification in the manuscript on page 4. And the studies *Nano Lett.* 2018, 18, 4, 2498-2504, *Adv. Mater.* 2020, 32, 1907453, and *Adv. Mater.* 2021, 33, 2104807 were cited in manuscript and added in reference [52-54].

(2) To substantiate the differences between localized and global melt transfers of gallium, we have conducted additional experiments to validate the efficiency of our PLMT by comparing localized and global molten transfer printing methods. We have included the added information addressing this clarification in the manuscript on page 7. The added experimental results are presented in Figure S3.

Our modification to the manuscript:

(1) On Page 4 of the main text, we added “On the basis of the phase change of metal gallium, it has reported a highly reversible and switchable adhesion, demonstrating the feasibility of using gallium's phase change for strong, reversible, and robust adhesion on various surfaces under different conditions. This approach has been effectively applied to macroscopic objects^[50]. Previous studies also provide critical insights into the behavior of gallium in transfer printing processes, such as the crystallization kinetics of gallium under varying thermal conditions, the liquid crystal structure of supercooled gallium, and the mechanical stability of gallium-oxide nanofilms encapsulating liquid gallium^[52-54]. Building on these findings, further research is needed to explore its application for transferring micro/nano-scale objects and delicate ultrathin films due to challenges such as difficult visual observation, small mass, and fragility of thin films. Additionally, completely melted gallium droplets pose operational challenges due to their high fluidity and tendency to easily fall. Therefore, a comprehensive analysis of the critical parameters influencing transfer accuracy and uniformity is essential. This includes further investigating the effects of the extent of gallium melting and the curvature of the melting stamp on the precision of the transfer printing process. Notably,

arrays transferred with the globally molten stamp show significant dislocations and extensive gallium residue. Addressing these challenges is imperative for achieving higher precision and cleaner transfers of ultrathin films when utilizing phase change materials, thereby extending the applicability of metal gallium for transfer printing”.

And on Page 5 of the main text, we added “Here in this work, we report a precision-induced (laser or hotplate) localized molten technique (PLMT) enabled by the Ga stamp (Figure 1A), which leverages the fluidity of localized molten liquid metal to avoid damage on fragile thin films and utilizes the solid part of the stamp to maintain structural integrity and improve operation reliability. Employing a nearly planar gallium stamp achieved through localized melting can significantly improve the transfer accuracy of thin film arrays and reduce gallium droplet residue compared to a fully molten droplet. Additionally, by adjusting laser power and spot size, the melting range, optimizing the transfer process can be precisely controlled. The simple yet robust thermal actuated, laser-transient-induced or hotplate-directional-induced local molten Ga stamp can provide gentle preload, intimate contact, and highly reversible adhesion strength.”

(2) On Page 7 of the main text, we added “The localized molten Ga stamp can also help to reduce the dislocation and the pollution of Ga stamp caused by the drop of liquid metal. To validate the efficiency of PLMT, we compared the differences between localized and global molten Ga stamp transfer printing methods. The experimental results are presented in Figure S3. A 2×2 square silicon platelet array (each with dimensions of $200 \mu\text{m} \times 200 \mu\text{m} \times 2 \mu\text{m}$) was initially picked up by a planar solid gallium metal with an infinite radius of curvature (Figure S3A(i)). Subsequently, the array was transferred using both a localized molten gallium stamp, which exhibits a slightly reduced radius of curvature compared to a perfectly planar surface and only undergoes surface flow (Figure S3A(ii)), and a global molten gallium stamp, which forms a single droplet with a radius of curvature decreasing to that of the liquid droplet ($\sim 2 \text{ mm}$) (Figure S3A(iii)). When printing the array onto a PDMS substrate using the localized molten stamp, the array remained highly regular, with minimal gallium residue around it (Figure S3B, right with a red solid line box). In contrast, the array transferred with the global molten stamp showed significant dislocations and extensive gallium residue (Figure S3B, left with a blue solid line box). Therefore, localized transfer printing achieves higher precision with less contamination compared to global transfer printing. The increased deformation occurs because a droplet with a smaller radius of curvature needs to transition from a curved surface to a flat substrate upon contact. This transition involves greater deformation compared to a localized molten stamp with a larger radius of curvature approaching flatness. Ideally, ensuring contact between a planar stamp and the substrate can maintain transfer precision and prevent

dislocation during the printing process.”

(3) We have added Figure S3 to the supplementary materials to demonstrate the differences between localized and global molten transfer printing methods.

Figure S3. Comparison of localized and global molten Ga stamp transfer printing methods. (A) Sequential images illustrating the transfer printing process of a 2×2 square silicon platelet array. (B) Images of the silicon platelet arrays printed on PDMS surfaces by global molten method (left with a blue solid line frame) and localized molten method (right with a red solid line frame).

4. The novelty claimed in this manuscript is the transfer of wafers at nanoscale heights. Given the existing research on gallium oxide film changes under pressure (Nano Lett. 2018, 18, 4, 2498-2504) and crystallization kinetics dependent on substrate type (Adv. Mater. 2020, 32, 1907453), demonstrate that these factors do not adversely affect the transfer of nanoscale sheets in terms of morphological changes and device performance.

Our replies:

Thank you for your valuable suggestion. To address this concern, we conducted further investigations into the effects of gallium oxide film changes under pressure and crystallization kinetics on the transfer printing of nanoscale sheets.

Referring to recent studies on the crystallization kinetics of gallium under varying conditions (*Adv. Mater.* 2020, 32, 1907453) and the mechanical stability of gallium-oxide nanofilms encapsulating liquid gallium (*Nano Lett.* 2018, 18, 4, 2498-2504), we learned that the silicon nanomembranes are subjected to deformation from the gallium oxide film during preload and crystallization. Therefore, we captured Atomic Force Microscope (AFM) images of the nanoscale Si thin films before and after transfer and analyzed their surface roughness. The results showed no significant differences in surface roughness, indicating that the transfer process does not impact the geometric morphology of the Si sheets. And We have tested the resistivities of the nanoscale Si thin films before and after transfer. The results showed no significant differences in Si performance. We have included these findings in Figure S16 to support our conclusions.

To study the underlying mechanisms, we established a molecular dynamics model to simulate the interaction between liquid gallium metal with a nano gallium oxide layer and a nanoscale silicon sheet. Our analysis showed that after compressing the gallium stamp by 40% strain, the maximum strain on the gallium oxide layer reached 50%, while the strain on the nanoscale silicon sheet remained at 0.1%, far below the silicon's fracture strain of 1% (*Science*, 2006, 311, 5758, 208-212). Additionally, we analyzed the effects of the crystallization process of gallium metal, from liquid to solid, on nanoscale silicon sheets. As gallium metal crystallizes, it transitions from a disordered state to an ordered crystalline structure, resulting in volume expansion and additional force on the silicon sheet. The maximum strain on the silicon sheet increased to 0.5% post-phase transition, but it remains well below the fracture strain of 1%. Our molecular dynamics model results indicate that although the silicon sheet is subjected to gallium oxide film deform during preload and volume expansion during crystallization, the strain induced on the nanoscale silicon sheet is minimal and does not affect its performance.

We have included the added information addressing this in the manuscript on page 16 and in Supplementary Materials on Page 5. The added experimental results and simulation results are presented in Figure S16 and Figure S17, respectively. The study *Adv. Mater.* 2020, 32, 1907453, *Nano Lett.* 2018, 18, 4, 2498-2504 and *Science*, 2006, 311, 5758, 208-212 were cited in manuscript and added in reference [52], [54] and [66]. The study *J. Phys.: Condens. Matter* 15 5649 (2023), *Phys. Rev. B* 95, 224103 (2017) and *Appl. Phys. Lett.* 122, 031602 (2023) were cited in Supplementary Materials and added in Supplementary reference [8-10].

Our modification to the manuscript:

(1) On Page 16 of the main text, we added “To further verify that the PLMT methods do not damage the 600 nm Si membranes, and consequently do not affect their

performance, we refer to recent studies on the crystallization kinetics of gallium under varying conditions^[52] and the mechanical stability of gallium-oxide nanofilms encapsulating liquid gallium^[54]. Based on these studies, we learned that the silicon nanomembranes are subjected to deformation from the gallium oxide film during preload and crystallization. Thus, Atomic Force Microscope (AFM, Cypher ES, Oxford Instruments Technology Co., Ltd) images of the nanoscale Si films were captured before (Figure 3B, i) and after (Figure 3B, ii) transfer print, and their surface roughness was analyzed. As shown in Figure S16A and 16B, there were no significant differences in surface roughness between before (RMS roughness lower than 0.96 nm) and after transfer printing (RMS roughness lower than 0.99 nm). In Figure 16C, the resistivity results of a Si nanomembrane are ~ 7.5 ohm-cm and ~ 7.6 ohm-cm before and after transfer printing processes, respectively. This indicates that the transfer process does not impact the surface morphology and performance of Si nanomembranes. To understand the underlying mechanisms, we utilized Molecular Dynamics (MD) simulations at the nanoscale to reflect the corresponding microscopic mechanical behavior in the dynamic contact process between the Ga-based stamp and the silicon nanomembrane. The utilized forcefield in MD simulations can accurately describe the interactions among atoms, and the collective behavior of atoms can represent the strained condition of the material system based on the principle of statistical mechanics. Therefore, we conducted the simulations to investigate the gallium oxide film changes under pressure and crystallization kinetics during the liquid-to-solid transition of gallium droplets in contact with silicon nanoscale sheets during the PLMT process, and their impact on the performance of nanoscale sheets. In Figure S17A, the simulations analyzed the local atomic strain distribution after compressing the gallium stamp by 40%, revealing that the strain accumulated mainly at the interface, simulation details in Supplementary Materials: Molecular dynamics (MD) simulation methods. The maximum strain on the gallium oxide layer reached approximately 50% (Figure S17B), while the maximum strain on the silicon wafer remained around 0.1% (Figure S17C, top), far below the silicon's fracture strain of 1%^[66]. Considering the volume expansion^[66] caused by the crystallization of gallium metal when it has already been compressed 40% from liquid to solid, the maximum strain on the silicon wafer increased to around 0.5% (Figure S17C, bottom), which is still below the fracture strain of 1%. The MD simulation results indicate that although the silicon sheet is subjected to deformation from the gallium oxide film during preload and volume expansion during crystallization, the induced strain on the nanoscale silicon sheet is minimal and does not affect its performance.”

(2) On Page 5 of the supplementary materials, we added Molecular dynamics (MD) simulation methods “Molecular dynamics (MD) simulations were performed to investigate the mechanical behaviors between the Gallium (Ga)-based stamp and the silicon wafer during the contact process. The Ga-based liquid metal contained two

components: the Gallium oxide (Ga_2O_3) film and the liquid Gallium core. During the modeling, the film model of Ga_2O_3 in spherical-shell structure was constructed with the diameter of 12 nm and the thickness of 2 nm. The Ga liquid metal with 8 nm diameter was added into the inner of the Ga_2O_3 shell to create the Ga-based stamp. The nanoscale crystalline silicon with the dimensions of $5.43 \times 5.43 \times 3.26 \text{ nm}^3$ and $\alpha = \beta = \gamma = 90^\circ$ was constructed as the wafer under the Ga-based stamp. The initial model was shown in Figure S15A. The Tersoff based interatomic potential for Ga liquid metal and silicon wafer, combined with a Born-Mayer-Huggins type expression for Ga_2O_3 film was applied to describe the interactions among metallic atoms in this system, which has been verified and evaluated for the related metallic systems^[8-10]. The energy minimization was performed to obtain a stable structure. The system was then equilibrated under the NVT ensemble (constant number of particles, volume and temperature) at a constant temperature of 300 K for 1 ns to achieve an equilibrium state. After full equilibrium, MD simulations for the contact process were performed by moving the Ga-based stamp towards the silicon wafer until the stamp was deformed with the strain of 40%. The non-periodic boundary conditions were applied in all three dimensions. To analyze the effect of the crystallization process of gallium metal, the Ga-based stamp was cooled down to 50 K when it has already been compressed 40% on the silicon wafer. The local atomic strain distribution with the reference to the relaxed system before the contact was captured to analyze the deformation status in the system moving during the contact process.”

(3) We added Figure S16 to illustrate the surface roughness and resistivity results of the nanoscale Si films captured before and after the transfer printing process, demonstrating that there are no significant differences in surface morphology and performance between the two states.

(4) We added Figure S17 to demonstrating the Molecular Dynamics (MD) simulations to investigate the gallium oxide film changes under pressure and crystallization kinetics during the liquid-to-solid transition of gallium droplets in contact with silicon nanoscale sheets during the PLMT process.

Figure S16. (A) Atomic force microscope images of a 600 nm Si membrane before (left) and after (right) the transfer print process. (B) The surface roughness data and the root-mean-square roughness values of samples before and after transfer print process. (C) The resistivity values of the Si nanomembrane before and after transfer print process.

Figure S17. (A) The atomistic structure and contact process of the Ga-based stamp on the nanoscale silicon wafer. (B) The local atomic strain distribution of the whole system. (C) The atomistic structure and contact process with the local atomic strain distribution when Ga was transformed from liquid state to solid state.

5. The manuscript describes the use of AZ 5214 photoresist on ultra-thin silicon wafers, where both gallium and the photoresist are in contact. Measure and report the adhesive strength between these materials, ensuring that the presence of AZ 5214 does not exacerbate contamination.

Our replies:

Thank you for your valuable feedback. We appreciate your attention to detail. In our experimental setup, we employed ultraviolet exposure to prepare masks from the photoresist, followed by inductively coupled plasma (ICP) etching to create Si square

pellets (400 μm by 400 μm) arranged in an array. Subsequently, any excess AZ 5214 photoresist was removed using acetone. It's worth noting that since the photoresist has been washed off with acetone, the liquid metal actually comes into direct contact with Si. We have added the information addressing this clarification in the Materials and Methods section, specifically on page 25.

Our modification to the manuscript:

On Page 25 of the main text, we added “Then, the remaining photoresist and exposed SiO_2 were removed by immersing in acetone for 5 mins and hydrofluoric acid (49%) for 2 mins, respectively.”

Reviewers' Comments and Our Responses

Reviewer #1:

The revised manuscript is now suitable for publication. Most issues have been adequately addressed.

Our replies:

We appreciate the reviewer's positive feedback and recognition of our work.

Reviewer #2:

The authors have reasonably addressed the reviewer comments. The manuscript is recommended for publication.

Our replies:

We appreciate the reviewer's positive feedback and recognition of our work.

Reviewer #3:

Reviewer #4:

The author has provided a response to my comments; however, I still find the research based on the localized Ga metal transfer via laser heating for the patterning of nm-thick Si layers to be lacking in appeal. The technique of localized heating for elastomer stamp transfer is already established (PNAS, 2024, 121(5), e2318739121), and there are studies that have demonstrated wafer-scale patterning with larger surface areas than the mm-scale pattern transfer conducted in this manuscript (Adv. Mater. Technol. 2020, 5, 2000549). Therefore, I do not support this study is suitable for publication in Nature Communications.

- There are already studies that have successfully transferred small chips as small as 4 μm in size (Nature, 2023, 614, 81–87). Please explain the minimum chip size that can be transferred using the technique presented in this manuscript. Additionally, compared to other studies that have transferred micro-scale chips, please elaborate on

the technical advantages of your approach.

- There are also studies that have performed large-area wafer-scale patterning (Adv. Mater. Technol. 2020, 5, 2000549). It would be important to provide an explanation of the maximum area that can be patterned using your technique.

Our replies:

We thank the reviewer for raising valuable comments. We have addressed the comments accordingly, and we believe that the questions will help improve our manuscript significantly.

(1) Thank you for your valuable suggestions. We will clarify in the introduction, abstract and discussion paragraph that the technique of localized heating for elastomer stamp transfer is already well-established, as well as acknowledge in the introduction that studies have demonstrated wafer-scale patterning with larger surface areas. We also appreciate the reviewer's reference to (PNAS, 2024, 121(5)), which is in fact one of our recently published works. This paper has indeed inspired our current study on liquid metal transfer.

During the transfer printing process using liquid gallium metal, we encountered challenges in controlling the droplet due to its high fluidity and tendency to fall easily, and we found that localized heating for elastomer stamp transfer provides precise control over the transfer process (PNAS, 2024, 121(5), e2318739121; National Science Review, 2020, 7, 296–304; ACS Appl. Mater. Interfaces 2016, 8, 35628–35633). Thus, we extended the elastomer-based localized molten technique to liquid metal. Interestingly, localized melting of gallium metal effectively improves the reliability of the transfer process. However, unlike the solid nature of elastomer stamps, the fluidity of liquid metals introduces challenges. This requires careful exploration of the feasibility of localized melting, including investigating the extent of gallium melting and the curvature of the molten stamp to ensure precision in the transfer printing process.

As the reviewer mentioned, the work on wafer-scale patterning with larger surface areas (Adv. Mater. Technol. 2020, 5, 2000549; Sci. Adv. 2020; 6: eabb2393) is highly significant. However, large-scale printing with liquid metal poses significant challenges due to its fluidity. A larger droplet could increase throughput but might cause the molten area to become more rounded, leading to some dislocation in the pattern. We are currently developing solutions to address these challenges, which will be presented in our future work.

(2) We appreciate the reviewer's insightful comment and the opportunity to clarify our work. In our current study, the smallest chip we have successfully transferred has a

minimum length and width of 200 μm , and a thickness of 600 nm. While we believe that our technique is theoretically capable of transferring even smaller length and width chips, our chip's size is currently limited by the manufacturing precision of our laboratory equipment. However, we would like to emphasize that our approach has been specifically designed to address the challenges associated with transferring ultra-thin devices using elastomeric stamps. In particular, we focus on preventing damage during the transfer of ultra-thin membranes, which can be especially prone to damage due to the contact forces applied during the pick-up and print stages. This is particularly significant when dealing with devices where the length-to-thickness ratio is large. From a technical perspective, transferring larger length and width membranes with the same thickness becomes progressively more difficult, as larger chips are more vulnerable to deformation or breakage under shear forces during the preloading process (Extreme Mechanics Letters 7 (2016) 136–144). This is why we believe our technique can handle smaller chips without damage, as other studies have a smaller length-to-thickness ratio compared to the objects we currently handle. Furthermore, by closely studying the forces involved in the elastomeric stamp transfer process, we have developed a technique that minimizes the risk of damage, making it highly suitable for the transfer of delicate, thin films that are challenging to handle using traditional techniques. We still appreciate your suggestion, and in our future work, we will strive to transfer even smaller-scale chips.

(3) We acknowledge the reviewer's concern regarding large-area wafer-scale patterning. Due to the inherent fluidity of liquid metal, we have currently demonstrated arrays with fewer than ten elements. We have addressed this challenge on page 19 of the manuscript: "Moreover, the proposed process is not compatible with large-scale printing, in which a larger droplet would increase throughput but bring the localized molten area more round making the pattern dislocate to some extent." However, we are actively working on a solution to enable large-scale printing with liquid gallium, which we plan to discuss in an upcoming publication.

Our modification to the manuscript:

(1) On Page 1 of the Abstract, we added "Transfer printing, a crucial technique for heterogeneous integration, has gained attention for enabling unconventional layouts and high-performance electronic systems. Elastomer stamps are typically used for transfer printing, where localized heating for elastomer stamp can effectively control the transfer process. A key challenge is the potential damage to ultrathin membranes from the contact force of elastic stamps, especially with fragile inorganic nanomembranes."

(2) On Page 5 of the Introduction, we added "Additionally, completely melted gallium droplets pose operational challenges due to their high fluidity and tendency to

easily fall. Therefore, a comprehensive analysis of the critical parameters influencing transfer accuracy and uniformity is essential. Inspired by the precision control achieved through localized heating in elastic stamp transfer printing^[31-33], which allows for controllable deformation, we adapted this technique for use with liquid metals. Unlike elastomer stamps, the fluid nature of liquid metals introduces challenges that require careful investigation of localized melting. This includes further investigating the effects of the extent of gallium melting and the curvature of the melting stamp on the precision of the transfer printing process.” And added “Notably, previous studies have shown that elastomer stamps can achieve wafer-scale patterning over larger surface areas^[19,47]. However, arrays transferred using a globally molten stamp exhibit significant dislocations and extensive gallium residue due to the fluidity of the molten metal.”

(3) On Page 22 of the discussion paragraph, we added “These characteristics are realized by localized molten metal Ga through precision-induced external thermal stimulus. While localized heating for elastomer stamp transfer is an already well-established technique, the proposed localized molten stamp features a gentle contact force and exceptional conformal adaptability and yet improves operation reliability when compared to the fully liquefied Ga stamp.”

(4) The studies *PNAS*, 2024, 121(5) and *Adv. Mater. Technol.* 2020, 5, 2000549 were cited in the manuscript and added in reference [33] and [47], respectively.